# Tanimoto Random Features for Scalable Molecular Machine Learning

**Austin Tripp**
Unversity of Cambridge
ajt212@cam.ac.uk

**Sergio Bacallado**
Unversity of Cambridge
sb2116@cam.ac.uk

**Sukriti Singh**
University of Cambridge
ss2971@cam.ac.uk

**José Miguel Hernández-Lobato**
University of Cambridge
jmh233@cam.ac.uk

## Abstract

The Tanimoto coefficient is commonly used to measure the similarity between molecules represented as discrete fingerprints, either as a distance metric or a positive definite kernel. While many kernel methods can be accelerated using random feature approximations, at present there is a lack of such approximations for the Tanimoto kernel. In this paper we propose two kinds of novel random features to allow this kernel to scale to large datasets, and in the process discover a novel extension of the kernel to real-valued vectors. We theoretically characterize these random features, and provide error bounds on the spectral norm of the Gram matrix. Experimentally, we show that these random features are effective at approximating the Tanimoto coefficient of real-world datasets and are useful for molecular property prediction and optimization tasks. Future updates to this work will be available at http://arxiv.org/abs/2306.14809.

## 1 Introduction

In recent years there have been notable advances in the use of machine learning (ML) for drug discovery, including molecule generation and property prediction (Dara et al., 2022). Despite ceaseless progress in deep learning, conventional methods such as support vector machines or random forest trained on *molecular fingerprints* are still competitive in the low-data regime (Walters and Barzilay, 2020; Stanley et al., 2021). These fingerprints essentially encode fragments from a molecule into a sparse vector, thereby compactly representing a large number of molecular substructures (David et al., 2020). They are extensively used in virtual screening for substructure and similarity searches as well as an input for ML models (Cereto-Massagué et al., 2015; Granda et al., 2018).

The *Tanimoto coefficient* (also known as the *Jaccard index*) stands out as a natural way to compare such fingerprints. This coefficient is most commonly expressed as a function on sets $T_S$ or as a function of non-negative vectors $T_{MM}$ (Jaccard, 1912; Tanimoto, 1958; Ralaivola et al., 2005; Costa, 2021; Tan et al., 2016):

$$T_S(X, X') = \frac{|X \cap X'|}{|X \cup X'|}, \ \ X, X' \subseteq \Omega, \qquad T_{MM}(x, x') = \frac{\sum_i \min(x_i, x'_i)}{\sum_i \max(x_i, x'_i)}, \ \ x, x' \in \mathbb{R}^d_{\geq 0}. \quad (1)$$

If $a$ and $b$ are binary indicator vectors representing sets $A$ and $B$ respectively, then $T_S(A, B) = T_{MM}(a, b)$. Therefore $T_{MM}$ can be viewed as a generalization of $T_S$; for this reason it is sometimes called the "weighted Jaccard coefficient" or min-max coefficient.

37th Conference on Neural Information Processing Systems (NeurIPS 2023).

The Tanimoto coefficient is widely used in machine learning and cheminformatics to compute similarities between molecular fingerprints (Bajusz et al., 2015; O'Boyle and Sayle, 2016; Miranda-Quintana et al., 2021), chiefly because of the following properties:

1. **Clear Interpretation:** The value of $T_{MM}(x, x')$ represents the degree of overlap between $x$ and $x'$ and is always between 0 and 1. $T(x, x') = 1$ only when $x = x'$.

2. **Kernel:** $T_{MM}(\cdot, \cdot)$ is positive definite (Gower, 1971; Ralaivola et al., 2005), meaning it can be used as the kernel for algorithms like support vector machines or Gaussian processes.

3. **Metric:** $1 - T_{MM}(x, x')$ is a valid distance metric (typically called *Jaccard/Soergel distance*) and can therefore be used in nearest-neighbour and clustering algorithms (Marczewski and Steinhaus, 1958; Levandowsky and Winter, 1971).

In this paper, we present and characterize two efficient low-rank approximations for large matrices of Tanimoto coefficients. The first method, presented in section 3, uses a random hash function to index a random tensor and enjoys exceptionally low variance. The second method, presented in section 4, uses a power series expansion of the Tanimoto similarity for binary vectors. This line of research also unexpectedly led to the discovery of a new generalization of the Tanimoto coefficient to arbitrary vectors in $\mathbb{R}^d$, $T_{DP}$, which is also a kernel and can be used to form a distance metric. In section 6 we demonstrate experimentally that our random features are effective at approximating Tanimoto matrices of real-world fingerprint data and demonstrate its application to molecular property prediction and optimization problems.

## 2  Background: kernel methods and random features

Kernel methods are a broad class of machine learning algorithms which make predictions using a positive definite *kernel function* $k : \mathcal{X} \times \mathcal{X} \mapsto \mathbb{R}$ (Schölkopf et al., 2002). Common methods in this class are support vector machines (Cortes and Vapnik, 1995) and Gaussian processes (Williams and Rasmussen, 2006). Given a dataset of $n$ data points, training most kernel methods requires computing the $n \times n$ kernel matrix[1] $K_{i,j} = k(x^{(i)}, x^{(j)})$ (with $O(n^2)$ time complexity) and possibly inverting it (with $O(n^3)$ time complexity). Because of this, applying kernel methods to large datasets generally requires approximations.

Given a kernel $k$, a *random features map* is a random function $f : \mathcal{X} \mapsto \mathbb{R}^M$, with the property that $f(x) \cdot f(x')$ approximates $k(x, x')$ for every pair $x, x' \in \mathcal{X}$. The approximation is often exact in expectation:

$$\mathbb{E}_f [f(x) \cdot f(x')] = k(x, x') \quad \text{for all } x, x' \in \mathcal{X}. \tag{2}$$

Random features allow the kernel matrix to be approximated as $\widehat{K}_{i,j} = f(x^{(i)}) \cdot f(x^{(j)})$. Because this matrix has rank at most $M$, this approximation generally reduces the cost of $O(n^3)/O(n^2)$ computations to $O(M^3)/O(nM^2)$, i.e. at most linear in $n$.

The seminal work of Rahimi and Recht (2007), which coined the term random features, gave a general method based on Fourier analysis to construct random features for any *Bochner* or *stationary kernel*, for which $k(x, x')$ is a function of $x - x'$. This class includes many common kernels including the RBF and Matérn kernels, but excludes $T_{MM}$. Subsequent works have proposed random features for other kernels including the polynomial kernel and the arc-cosine kernel (Liu et al., 2021). However, there is no general formula to define random features for non-stationary kernels, such as $T_{MM}$.

A random features map is sometimes called a *data-oblivious sketch*, to distinguish it from other *data-dependent* low-rank approximation methods which depend on a given dataset $x^{(1)}, \ldots, x^{(n)}$. Examples of data-dependent low rank sketches are the Nyström approximation and leverage-score sampling (Drineas et al., 2005, 2012). Although data-dependent methods may result in lower approximation errors for a given dataset, data-oblivious sketches are naturally parallelizable and useful in cases where the dataset changes over time (e.g. streaming or optimization) or for ultra-large datasets which may not fit in memory.

---

[1] We write $x_i$ to refer to the $i$th element of a vector and $x^{(i)}$ to denote the $i$th vector in a list.

## 3 Low-variance random features for Tanimoto and MinMax kernels

Outside of chemistry, the Tanimoto coefficient has been widely used to measure the similarity between text documents and rank results in search engines. To quickly find documents with high similarity to a user's query, many prior works have studied random *hashes* for the Tanimoto coefficient, i.e. a family of random functions $h : \mathcal{X} \mapsto \{1, \ldots, K\}$ such that

$$\mathbb{P}_h \left( h(x) = h(x') \right) = T_{MM}(x, x'). \tag{3}$$

Although initially these hashes were only applicable to binary inputs (Broder, 1997; Broder et al., 1998; Charikar, 2002), more recent work has produced efficient random hashes for arbitrary non-negative vectors (Manasse et al., 2010; Ioffe, 2010; Shrivastava, 2016). In this section we propose a novel family of low-variance random features for $T_{MM}$ (and by extension $T_S$) which is based on random hashes.

It is important to clarify that although the definition of random hashes in equation 3 resembles the definition of random features in equation 2, they are actually distinct. Random hash functions output *discrete* objects (typically an integer or tuple of integers) whose probability of *equality* is $T_{MM}$, while random features must output *vectors* in $\mathbb{R}^M$ whose expected *inner product* is $T_{MM}$. If a random hash maps to $\{1, \ldots, K\}$, a naive approach may be to use a $K$-dimensional indicator vector as a random feature. Because hash equality is a binary outcome, the variance of such random features would be $T_{MM}(1 - T_{MM})$. Realistic hash functions like that of Ioffe (2010) use $K \geq 10^3$, implying a "variance per feature" of $\approx 10^2$, which is undesirably high.

Our main insight is that low-variance scalar random features can be created by using a random hash to *index* a suitably distributed random vector. In the following theorem, we show that a vector of i.i.d. samples from any distribution with the correct first and second moments can be combined with random hashes to produce random features for $T_{MM}$.

**Theorem 3.1.** *Let* $h : \mathcal{X} \to \mathcal{Y}$ *be a random hash for* $T_{MM}$ *satisfying equation 3, with* $|\mathcal{Y}| = K$. *Furthermore, let* $\xi$ *be a random variable such that* $\mathbb{E}[\xi] = 0$ *and* $\mathbb{E}[\xi^2] = 1$, *and let* $\Xi = [\xi_1, \ldots, \xi_K]$ *be a vector of independent copies of* $\xi$. *Then the 1D random features*

$$\phi_{\Xi,h}(x) = \Xi_{h(x)} \tag{4}$$

*estimate* $T_{MM}$ *without bias:* $\mathbb{E}_{\Xi,h}(\phi_{\Xi,h}(x) \cdot \phi_{\Xi,h}(x')) = T_{MM}(x, x')$, *and with variance*

$$\mathbb{V}_{\Xi,h} \left[ \phi_{\Xi,h}(x) \cdot \phi_{\Xi,h}(x') \right] = 1 + T_{MM}(x, x') \left( E[\xi^4] - 1 - T_{MM}(x, x') \right) \geq 1 - T_{MM}(x, x')^2. \tag{5}$$

*Furthermore, the lower bound is tight and achieved when* $\xi$ *is Rademacher distributed (i.e. uniform in* $\{-1, 1\}$).

The proof is given in Appendix D.1. This theorem shows that Rademacher $\xi$ yields the smallest possible variance in the class of random features defined in eq. (4).

These random features have many desirable properties. First, unlike random features for many other kernels such as the Gaussian kernel (Liu et al., 2021), the variance does not depend on the dimension of the input data or norms of the input vectors. Second, because these random features are 1-dimensional scalars, $M$ independent random feature functions can be concatenated to produce $M$-dimensional random feature vectors with variance at most $1/M$. This suggests that as few as $\approx 10^3$ random features could be used in practical problems. Third, although each instance of $\Xi$ requires storing a $K$ dimensional random vector, if $\xi$ is chosen to be Rademacher distributed, then each entry can be stored with a single bit, requiring just $\approx 100$ kB of memory when $K = 10^6$.

One disadvantage of these random features is that they are not continuous or differentiable with respect to their inputs. For applications such as Bayesian optimization which require optimizing over model inputs this would create difficulties as gradient-based optimization could no longer be done. It was this disadvantage which motivated us to search for other random features, leading to the discoveries in the following section.

## 4 Tanimoto dot product kernel and its random features

Ralaivola et al. (2005) gave a definition for the Tanimoto coefficient involving dot products:

$$T_{DP}(x, x') = \frac{x \cdot x'}{\|x\|^2 + \|x'\|^2 - x \cdot x'}, \tag{6}$$

with $T_{DP}(x, x') = 1$ when $x, x' = 0$. It is easy to check that $T_{DP}(x, x') = T_{MM}(x, x')$ on binary vectors, which was used by Ralaivola et al. (2005) to prove that $T_{DP}$ is a kernel on the space $\{0, 1\}^d$, referencing prior work by Gower (1971). However, $T_{DP}$ is not identical to $T_{MM}$ for general inputs $x, x' \in \mathbb{R}_{\geq 0}^d$. Here, we give the first proof that $T_{DP}$ is a positive definite function in $\mathbb{R}^d$ and thus, also a valid kernel in this space.

**Theorem 4.1.** *For $x, x' \neq 0$ in $\mathbb{R}^d$, we have*

$$T_{DP}(x, x') = \sum_{r=1}^{\infty} (x \cdot x')^r \left( \|x\|^2 + \|x'\|^2 \right)^{-r}, \tag{7}$$

*where the series is absolutely convergent. The function $T_{DP}$ is a positive definite kernel in $\mathbb{R}^d$.*

It has been noticed previously that, unlike $1 - T_{MM}$, the function $1 - T_{DP}$ is *not* a distance metric on non-binary inputs (Kosub, 2019). Indeed, when $d = 1$, the inputs $\{1, 2, 4\}$ violate the triangle inequality. However, we can easily derive a distance metric from $T_{DP}$.

**Corollary 4.2.** $d_{DP}(x, x') = \sqrt{1 - T_{DP}(x, x')}$ *corresponds to the RKHS norm of the function* $\frac{1}{2}[T_{DP}(x, \cdot) - T_{DP}(x', \cdot)]$ *and is therefore a valid distance metric on $\mathbb{R}^d$.*

Proofs are given in Appendix D.2. These results imply that $T_{DP}$, like $T_{MM}$, is an extension of the set-valued Tanimoto coefficient (equation 1) to real vectors and can be used as a substitute for $T_{MM}$ in machine learning algorithms that require a kernel or distance metric. Unlike $T_{MM}$, the kernel $T_{DP}$ is differentiable everywhere with respect to its inputs. It can also be computed in batches using matrix-matrix multiplication, allowing for efficient vectorized computation.

We now consider producing a random features approximation to $T_{DP}$ for large-scale applications. Motivated by the close relationship between $T_{DP}$ and $T_{MM}$, one may be tempted to find a random hash for $T_{DP}$ and apply the techniques developed in section 3. Unfortunately, we are able to prove that this is not possible.

**Proposition 4.3.** *There exists no random hash function for $T_{DP}$ over non-binary vectors.*

*Proof.* Charikar (2002) proved that if $s(x, x')$ is a similarity function for which there exists a random hash, then $1 - s(x, x')$ must satisfy the triangle inequality (see their Lemma 1). Because $1 - T_{DP}(x, x')$ does not satisfy the triangle inequality it follows by contradiction that there does not exist a random hash for $T_{DP}$. $\qquad \square$

Therefore producing random features for $T_{DP}$ will require another approach. In the remainder of this section we present a framework to produce random features for $T_{DP}$ by directly approximating its power series (equation 7). We first describe a method to produce random features for $(\|x\|^2 + \|x'\|^2)^{-r}$ (4.1). Then we describe how these features can be combined with existing random features for the polynomial kernel to approximate $T_{DP}$'s truncated power series (4.2–4.3). Lastly, we present an error bound for the kernel matrix of a dataset in the spectral norm, showing that the required dimension for the sketch scales optimally with the stable rank of the kernel matrix (4.4).

## 4.1 Random features for the "prefactor" $\left( \|x\|^2 + \|x'\|^2 \right)^{-r}$

In this section we present a random feature map for the positive definite kernel $(x, x') \mapsto (\|x\|^2 + \|x'\|^2)^{-r}$, which we will refer to as the the *prefactor*. We defer all proofs to Appendix D.3. We begin with the following lemma, which defines scalar random features for the prefactor:

**Lemma 4.4.** *If $Z \sim \mathrm{Gamma}(s, c)$ (where $c$ is a* rate *parameter), then*

$$\varphi_{r,Z}(x) = e^{(1/2 - \|x\|^2)Z} Z^{(r-s)/2} \sqrt{c^{-s} e^{(c-1)Z} \Gamma(s)/\Gamma(r)} \tag{8}$$

*is an unbiased scalar random feature for the prefactor $(\|x\|^2 + \|x'\|^2)^{-r}$ for all $s, c > 0$.*

Although independent copies of $Z$ could be combined to form an $M$-dimensional sketch, we instead propose to use a dependent point set $Z_1, \ldots, Z_M$ where each element $Z_i$ has a $\mathrm{Gamma}(s, c)$ distribution whilst maximally covering the real line. This is a well-established Quasi-Monte Carlo (QMC) technique which generally attains lower variance. We define our $M$-dimensional QMC features in the following lemma:

**Lemma 4.5.** *Let $\gamma_{s,c}$ be the inverse cumulative distribution function of a* $\mathrm{Gamma}(s,c)$ *random variable. Fix $M, r \in \mathbb{N}$, $u \in (0,1)$, $c, s > 0$ and let $u_i = u + i/M - \lfloor u + i/M \rfloor$ for $i = 1, \ldots, M$. Define $\phi_{u,r}(x) = (\phi_{u,r,1}(x), \ldots, \phi_{u,r,M}(x))$, where:*

$$\phi_{u,r,i}(x) = \frac{1}{\sqrt{M}} \sqrt{\frac{c^{-s}\Gamma(s)}{\Gamma(r)}} e^{-(\|x\|^2 - c/2)\gamma_{s,c}(u_i)} (\gamma_{s,c}(u_i))^{(r-s)/2} . \tag{9}$$

*If $u \sim \mathcal{U}(0,1)$ then $\phi_{u,r}(x)$ forms unbiased random features of the prefactor $(\|x\|^2 + \|x'\|^2)^{-r}$.*

Although the random features are unbiased for all $s, c > 0$, the value of these parameters will impact the error. We show that if $s, c$ are suitably tuned, then the relative error can be bounded:

**Lemma 4.6.** *Let $x^{(1)}, \ldots, x^{(n)} \in \mathbb{R}^d$ with $\frac{\min_i \|x^{(i)}\|^2}{\max_i \|x^{(i)}\|^2} \geq \zeta$, and fix $u \in [0,1]$. Define the relative error*

$$E_{i,j} = \frac{\phi_{u,r}(x^{(i)}) \cdot \phi_{u,r}(x^{(j)}) - (\|x^{(i)}\|^2 + \|x^{(j)}\|^2)^{-r}}{(\|x^{(i)}\|^2 + \|x^{(j)}\|^2)^{-r}}. \tag{10}$$

*If $c = 2\zeta^2$, $s = r\zeta$, then for some constant $C$ independent of $r$ this error satisfies,*

$$\max_{1 \leq i,j \leq n} |E_{i,j}| \leq \frac{2}{M} \frac{\Gamma(r\zeta)\zeta^{-r\zeta}}{\Gamma(r)} (r/e)^{r(\zeta-1)}(1.3)^r \leq C(M\zeta)^{-1}. \tag{11}$$

Together, these lemmas suggest random features for the prefactor can be created by first estimating $\zeta$ (the minimum ratio of norms of input vectors), then using the random features from Lemma 4.5 with the values of $s, c$ specified in Lemma 4.6.

### 4.2 A framework to produce random features for $T_{DP}$

There are straightforward rules for producing random features for sums and products of kernels whose individual random features are known (Duvenaud, 2014, sec. 2.6.2). Random features for kernels $k_1, k_2$ can be *concatenated* (denoted $\oplus$) to form random features for the sum kernel $k_1 + k_2$, while their *tensor product*[2] (denoted $\otimes$) forms random features for the product kernel $k_1 \times k_2$. Our strategy to produce features for $T_{DP}$ is to combine random features for the "prefactor" (presented in section 4.1) with random features for the polynomial kernel to produce random features for $T_{DP}$'s power series (equation 7) truncated at $R$ terms.

Fix $R \in \mathbb{N}$, and for $r = 1, \ldots, R$, let $\phi_r$ be a $m_r$-dimensional random features map for the prefactor $(\|x\|^2 + \|x'\|^2)^{-r}$ and let $\psi_r$ be a $m'_r$-dimensional random features map for $(x \cdot x')^r$. The function:

$$\tilde{\Phi}_R(x) = \oplus_{r=1}^R [\phi_r(x) \otimes \psi_r(x)] \tag{12}$$

is therefore a random feature estimate for $T_{DP}$'s power series, truncated at $R$ terms. Unfortunately, these random features have dimension $M = \sum_{r=1}^R m_r m'_r$ which depends on the *product* of the random features dimension of $\phi_r$ and $\psi_r$. Furthermore, the dimension $m'_r$ of the random features $\psi_r$ required to approximate the polynomial kernel $(x \cdot x')^r$ with good accuracy can scale poorly with $r$. For even modest values of $m_r, m'_r$ the resulting value of $M$ will likely be prohibitively large.

To remedy this, we turn to recent works which propose powerful linear maps to approximate tensor products with a lower-dimensional vector. Assuming $x^{(1)}, y^{(1)} \in \mathbb{R}^{d_1}$ and $x^{(2)}, y^{(2)} \in \mathbb{R}^{d_2}$, these maps are effectively random matrices $\Pi \in \mathbb{R}^{m \times (d_1 d_2)}$, which exhibit a *subspace embedding property* whereby $[\Pi(x^{(1)} \otimes x^{(2)})] \cdot [\Pi(y^{(1)} \otimes y^{(2)})]$ concentrates sharply around $(x^{(1)} \otimes x^{(2)}) \cdot (y^{(1)} \otimes y^{(2)})$. Critically, the product $\Pi(x^{(1)} \otimes x^{(2)})$ can be computed *without* instantiating either matrix $\Pi$ or the tensor product $x^{(1)} \otimes x^{(2)}$. Examples of such methods include TENSORSKETCH and TENSORSRHT (Pagh, 2013; Pham and Pagh, 2013; Ahle et al., 2020), but for generality we will simply refer to these methods as SKETCH. Defining a series of sketches $\mathrm{SKETCH}_r : \mathbb{R}^{m_r} \times \mathbb{R}^{m'_r} \mapsto \mathbb{R}^{\tilde{m}_r}$ we can modify $\tilde{\Phi}_R$ from equation 12 into:

$$\Phi_R(x) = \oplus_{r=1}^R \mathrm{SKETCH}_r [\phi_r(x), \psi_r(x)] \tag{13}$$

---

[2] For two vectors $x \in \mathbb{R}^{d_1}$ and $y \in \mathbb{R}^{d_2}$, define the tensor product $x \otimes y = \mathrm{vec}(xy^T) \in \mathbb{R}^{d_1 d_2}$.

which has output dimension $M = \sum_{r=1}^{R} \tilde{m}_r$, i.e. without any pathological dependencies on the dimensions of $\phi_r(x), \psi_r(x)$. However, because these features approximate a *truncated* power series, they will be biased downward due to the monotonicity of the power series (7). We propose two bias correction techniques to potentially improve empirical accuracy. One approach is to normalize the random features such that $\Phi(x) \cdot \Phi(x) = 1 = T_{DP}(x, x)$ for all $x \in \mathbb{R}^d$, i.e., such that the diagonal entries of the kernel matrix $K$ are estimated exactly. A second approach is based on sketching the residual of the power series. These approaches are presented in detail in Appendix E.

## 4.3 Implementing the random features

Instantiating the random features from the previous subsection (13) requires making concrete choices for $R$, $\text{SKETCH}_r, m_r, m_r', \tilde{m}_r, \phi_r, \psi_r$ for all $r$, and choosing a bias correction technique. There are many reasonable choices for $\text{SKETCH}_r$, such as $\text{TENSORSKETCH}$ (Pham and Pagh, 2013) and $\text{TENSORSRHT}$ (Ahle et al., 2020). These sketches generally allow $m_r, m_r', \tilde{m}_r$ to be chosen freely (although naturally error will increase as $\tilde{m}_r$ decreases). Many of these sketches can also be used as random features for the polynomial kernel $\psi_r$ (Wacker et al., 2022), either directly or as part of more complex algorithms like $\text{TREESKETCH}$ (Ahle et al., 2020) or complex-to-real sketches (Wacker et al., 2023). The QMC random features from section 4.1 can be used for the prefactor $\phi_r$, with the parameters $s, c$ chosen based on the anticipated norms of the input vectors.

The only remaining inputs are $R$ (the number of power series terms to approximate) and $\tilde{m}_1, \ldots, \tilde{m}_R$ (how many random features to use for each term). Assuming a fixed dimension $M$ for the final random features, this choice involves a bias variance trade-off, as a higher value of $R$ will reduce bias but require each term in the power series to have fewer features, thereby increasing variance. Intuitively, because the terms of the power series decrease monotonically the variance of terms for small $r$ is likely to dominate the overall variance, and therefore we surmise that a *decreasing* sequence for $\{\tilde{m}_r\}_{r=1}^{R}$ will be the best choice. Ultimately however we do not have theoretical results to dictate this choice in practice. We will evaluate these choices empirically in section 6.

## 4.4 Asymptotic error bound for $T_{DP}$ random features

Because many kernel methods use kernel matrices as linear operators, it is natural to examine the error of kernel approximations in the operator norm. Previous works have produced asymptotic error bounds of the random feature dimension $m$ required to achieve a relative approximation error of $\varepsilon$ in the operator norm. Defining $\tilde{sr}(K) = \text{Tr}(K)/\|K\|_{\text{op}}$, Cohen et al. (2015) show that $m = \tilde{\Omega}(\tilde{sr}(K)/\varepsilon^2)$ is essentially optimal for data-oblivious random features of linear kernels, even though it is possible to eliminate logarithmic factors. Our main theoretical result is that with the correct choices of base random features, the random features for $T_{DP}$ presented in section 4.2 achieve similar scaling. We now state this as a theorem.

**Theorem 4.7.** *For any $n \geq 1$, let $x^{(1)}, \ldots, x^{(n)} \in \mathbb{R}^d$ be a set of inputs with $\frac{\min_i \|x^{(i)}\|^2}{\max_i \|x^{(i)}\|^2} \geq \zeta$. Let $K$ be the matrix with entries $K_{i,j} = T_{DP}(x^{(i)}, x^{(j)})$. For all $\varepsilon > 0$, there exists an oblivious sketch $\Phi : \mathbb{R}^d \to \mathbb{R}^m$ with $m = \tilde{\Omega}(\tilde{sr}(K)/\varepsilon^2)$, such that*

$$\mathbb{P}_{\Phi}\big(\|\widehat{K} - K\|_{op} \geq \varepsilon \|K\|_{op}\big) \leq \frac{1}{poly(n)} \tag{14}$$

*where $\widehat{K}_{i,j} = \Phi(x^{(i)}) \cdot \Phi(x^{(j)})$. Furthermore, the sketch can be computed in time $\tilde{O}(\tilde{sr}(K)n\varepsilon^{-2} + nnz(X)\varepsilon^{-2} + n\zeta^{-1}\varepsilon^{-3})$.*

The random features in the theorem follow equation 13, with specific choices for $M$, $R$, and $\{\text{SKETCH}_r, \tilde{m}_r, \phi_r, \psi_r\}_{r=1}^{R}$ given in the proof in Appendix D.4. Theorem 4.7 essentially suggests that, with the correct settings, the error of the random features proposed in section 4.2 scales as well as one could reasonably expect for a kernel of this type. We would highlight that the computational cost of the sketch is sub-quadratic in $n$, and compares favourably with the cost of data-dependent low-rank approximation methods.

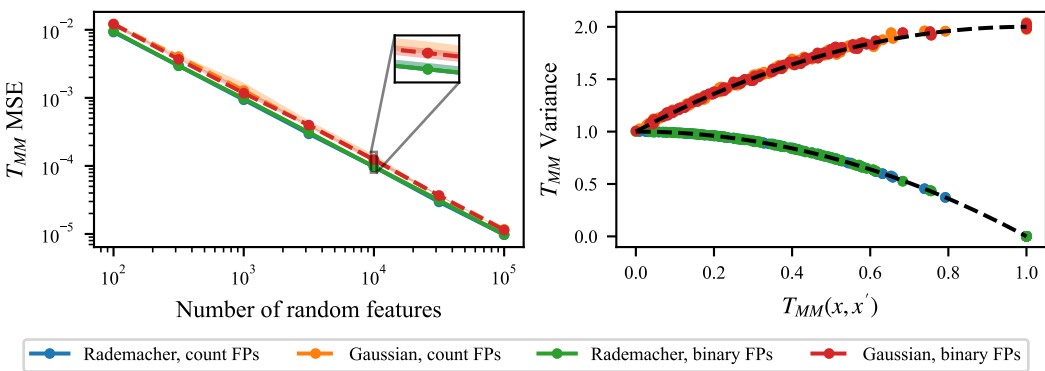

Figure 1: **Left:** MSE of $T_{MM}$ matrix reconstruction as a function of number of random features (median over 5 trials, shaded regions are first/third quartiles). **Right:** empirical variance of scalar $T_{MM}$ random feature estimates for $M = 10^5$, closely matching theoretical predictions (dashed lines).

## 5  Related work

Our work on random features fits into a large body of literature random features for kernels (Liu et al., 2021). The majority of work in this area focuses on stationary kernels (i.e. $k(x, x') = f(x - x')$), because the Fourier transform can be applied to any stationary kernel to produce random features in a systematic way (Rahimi and Recht, 2007). There is however no analogous universal formula to produce random features for non-stationary kernels like $T_{MM}$ and $T_{DP}$; therefore each kernel requires a bespoke approach. Although our random features are novel, they build upon ideas present in prior works. Our random features for $T_{MM}$ critically rely on previously-proposed random hashes for $T_{MM}$. Our approach to create random features for $T_{DP}$ via approximating its power series follows was inspired by the random features for the Gaussian kernel from Cotter et al. (2011), which were subsequently improved upon by Ahle et al. (2020). Similar techniques have also been used to create random features for the neural tangent kernel (Zandieh et al., 2021). However, to the best of our knowledge no prior works have proposed random features specifically for the Tanimoto kernel or its variants.

Other works have proposed other types of scalable approximations for Tanimoto coefficients which are not based on random features. Haque and Pande (2010) propose SCISSORS, an optimization-based approach to estimate Tanimoto coefficients which is akin to a data-dependent sketch. A large number of works use hash-based techniques to find approximate nearest neighbours with the Tanimoto distance metric (Nasr et al., 2010; Kristensen et al., 2011; Tabei and Tsuda, 2011; Anastasiu and Karypis, 2017). Although these techniques are useful for information retrieval, unlike random features they cannot be used to directly scale kernel methods to larger datasets.

## 6  Experiments

In this section we apply the techniques in this paper to realistic datasets of molecular fingerprints. All experiments were performed in python using the `numpy` (Harris et al., 2020), `pytorch` (Paszke et al., 2019), `gpytorch` (Gardner et al., 2018), and `rdkit` (Landrum et al., 2023) packages. Molecules were represented with both binary (B) and count (C) Morgan fingerprints (Rogers and Hahn, 2010) of dimension 1024 (additional details in Appendix F.1). These vectors indicate the presence (B) or count (C) of different subgraphs in a molecule. Code to reproduce all experiments is available at: https://github.com/AustinT/tanimoto-random-features-neurips23.

### 6.1  Errors of random features on real datasets

Here we study the error of approximating matrices of Tanimoto coefficients using our random features, with the general goal of verifying the claims in sections 3–4 on a realistic dataset of molecules. We choose to study a sample of 1000 small organic molecules from the GuacaMol dataset (Brown et al., 2019; Mendez et al., 2019) which exemplify the types of molecules typically considered in drug discovery projects. We use both binary (B) and count (C) fingerprints of radius 2.

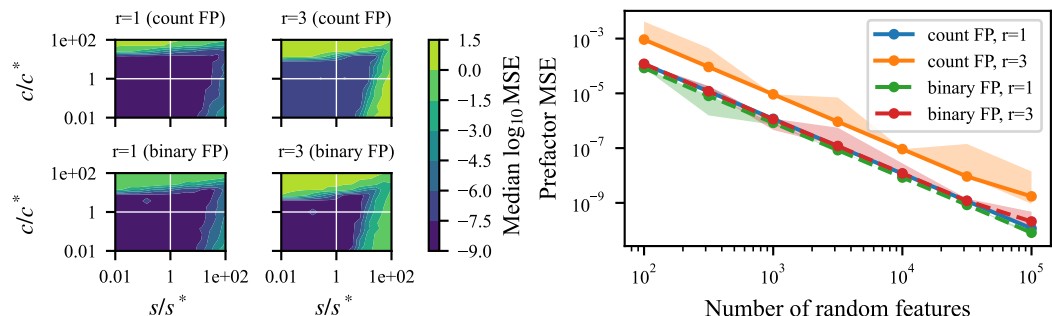

Figure 2: **Left:** Contour plots of MSE for prefactor random features with $M = 10^4$ with varying $s, c$. **Right:** MSE vs number of prefactor random features with $s, c$ values from Lemma 4.6. As in Figure 1, lines are medians over 5 trials, shaded regions are first/third quartiles.

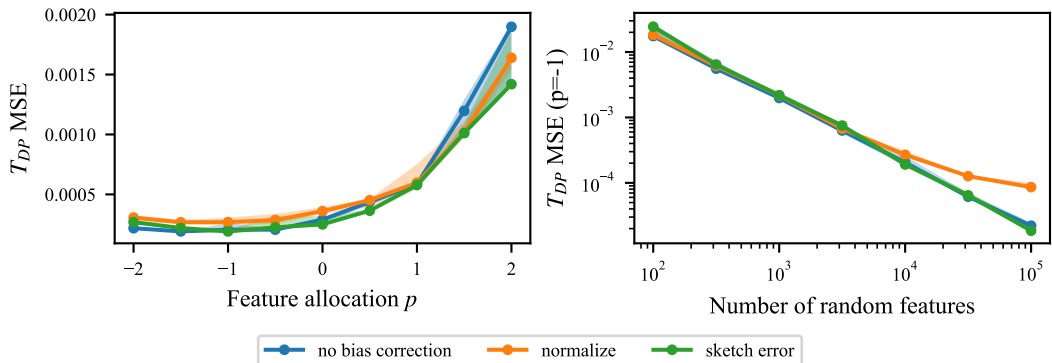

Figure 3: **Left:** MSE for $M = 10^4$ dimensional random features when allocating features by $\tilde{m}_r \propto r^p$, $r = 1 \ldots, 4$. **Right:** MSE of $T_{DP}$ random features using $R = 4, p = -1$ and various bias correction strategies. Both subplots use binary fingerprints (the equivalent plot for count fingerprints is Figure F.2). As in Figure 1, lines are medians over 5 trials, shaded regions are first/third quartiles.

First, we investigate the random features for $T_{MM}$ proposed in section 3. We instantiate these features using the random hash from Ioffe (2010) (explained further in Appendix F.2) with $\Xi$ both Gaussian and Rademacher distributed. The results are shown in Figure 1. The left subplot shows the median mean squared error (MSE), i.e. $\mathbb{E}_{i,j} \left[ \left( T_{MM}(x^{(i)}, x^{(j)}) - \phi(x^{(i)}) \cdot \phi(x^{(j)}) \right)^2 \right]$ as a function of the random feature dimension $M$. As expected for a Monte Carlo estimator, the square error decreases with $O(1/M)$ (i.e. increasing the number of random features by 10 reduces the MSE by a factor of 10). As predicted by Theorem 3.1, the estimation error seems to depend only on the distribution of $\Xi$ and not on the input vectors themselves; therefore the error curves for count and binary fingerprints overlap completely. The error is lowest when $\Xi$ is Rademacher distributed, although the empirical difference in error seems small. The right subplot looks at the variance across across scalar random features, showing close matching with the predictions of Theorem 3.1. Overall these features behave exactly as expected.

Next, we investigate the random features for $T_{DP}$ from section 4. These features are more complex, so we start by studying the random features for the "prefactor" from section 4.1. Recall that these features had free parameters $s, c > 0$. Fixing the number of features $M = 10^4$, Figure 2 (left) shows the MSE for the $r = 1$ and $r = 3$ terms for both the binary and count fingerprints (which have different norms) as a function of $s$ and $c$. The values which minimize the relative error bound from lemma 4.6, denoted $s^*, c^*$, seem to lie in a broad plateau of low error in all settings, suggesting that these values of $s, c$ are a prudent choice. Using these values of $s, c$, Figure 2 (right) shows the MSE with respect to the number of features $M$. As expected for a QMC method, the error dependence appears to be *quadratic* $O(1/M^2)$ (i.e. a ten-fold increase in $M$ reduces MSE by 100-fold).

Table 1: Average log probability of test set labels with various approximate GPs for 5 targets from DOCKSTRING dataset (García-Ortegón et al., 2022). $\pm$ values are standard deviations over 5 trials.

| KERNEL | METHOD | ESR2 | F2 | KIT | PARP1 | PGR |
|---|---|---|---|---|---|---|
| $T_{MM}$ | RAND SUBSET GP | -1.084±0.004 | -0.951±0.002 | -1.094±0.002 | -0.999±0.002 | -1.183±0.005 |
| | SVGP | -0.908±0.005 | -0.502±0.005 | -0.846±0.002 | -0.606±0.005 | -1.030±0.005 |
| | RFGP ($\Xi$ RAD.) | -0.954±0.009 | -0.658±0.013 | -1.222±0.044 | -0.968±0.037 | -1.127±0.023 |
| | RFGP ($\Xi$ GAUSS.) | -0.956±0.010 | -0.663±0.015 | -1.230±0.048 | -0.967±0.036 | -1.124±0.025 |
| $T_{DP}$ | RAND SUBSET GP | -1.073±0.002 | -0.940±0.001 | -1.077±0.002 | -0.988±0.001 | -1.187±0.006 |
| | SVGP | -0.880±0.004 | -0.459±0.002 | -0.804±0.002 | -0.568±0.002 | -1.010±0.004 |
| | RFGP (PLAIN) | -0.902±0.003 | -0.513±0.004 | -0.979±0.015 | -0.690±0.022 | -1.029±0.002 |
| | RFGP (NORM) | -0.902±0.003 | -0.515±0.003 | -0.980±0.015 | -0.691±0.022 | -1.028±0.002 |
| | RFGP (SKETCH) | -0.904±0.002 | -0.515±0.004 | -0.979±0.014 | -0.690±0.021 | -1.030±0.002 |

Because it is implemented in `scikit-learn` (Pedregosa et al., 2011), we use TENSORSKETCH (Pagh, 2013) both as the polynomial random feature map and to combine the polynomial and prefactor random features. We fix the number of random features for the prefactor to be $10^4$ (recall it can be chosen freely without impacting the final random feature dimension). Figure F.1 shows the MSE of approximating both $(x \cdot x')^r$ and $((x \cdot x')/(\|x\|^2 + \|x'\|^2))^r$. In both cases, the MSE decreases approximately with $O(1/M)$. Finally, we empirically examine how to allocate $M$ random features across $R$ terms. Using $R = 4$, Figure 3 (left) shows that allocating most of the features to the terms with small $R$ results in lower error. We therefore heuristically suggest allocating features according to $\tilde{m}_r \propto r^{-1}$. Recall that truncating the power series biases the random features downward, and in section 4.2 two bias correction techniques were proposed. Figure 3 (right) studies the overall MSE for the plain features and both bias correction techniques. It appears that, in practice, neither technique is particularly helpful (normalization in fact appears harmful for large $M$). All techniques show an error dependence of approximately $O(1/M)$.

## 6.2 Molecular property prediction and uncertainty quantification

To evaluate their efficacy in practice, we use our random features to approximate large-scale Gaussian processes (GPs) (Williams and Rasmussen, 2006) for molecular property prediction. Specifically, we study 5 tasks from the DOCKSTRING benchmark which entail predicting protein binding affinity from a molecular graph structure (García-Ortegón et al., 2022). Each task contains 250k molecules, making exact GP regression infeasible. We represent molecules with count fingerprints of radius 1.

We use $M = 5000$ random features for all methods. We compare to two approximate GP baselines. The first is an exact GP on a random subset of size $M$. Since this approach ignores most of the dataset, one should expect a reasonable approximate GP to perform better. The second is a sparse variational GP (SVGP) which approximates the dataset using $M$ pseudo-data points $Z$ (Titsias, 2009; Hensman et al., 2013). The locations of $Z$ are typically chosen based on the input dataset (we use K-means clustering), making this method effectively a data-dependent sketch. Accordingly, one might expect the performance of this approximation to be *better* than data-oblivious random features. Details of Gaussian process training are given in appendix F.4.

Table 1 shows the average log probability of test set labels for all types of GP with $T_{MM}$ and $T_{DP}$ kernels. Several trends are evident. First, for each kernel random feature GPs (RFGPs) consistently outperform random subset GPs, but underperform SVGP. Second, for each kernel the difference between the RFGP varieties is small (generally less than the standard deviation). Third, on most targets $T_{DP}$ seems to perform better than the $T_{MM}$ kernel. The reason for this is unclear. Similar trends can be see in the $R^2$ metric (Table F.1). This suggests that the RFGPs in this paper can be used in large-scale regression, although it seems in practice that data-dependent approximations are more accurate.

## 6.3 Bayesian optimization in molecule space via Thompson sampling

Bayesian optimization (BO) uses a probabilistic surrogate model to guide optimization and is generally considered one of the most promising techniques for sample-efficient optimization (Shahriari et al., 2015). Because wet-lab experiments are expensive and time-consuming, there is considerable interest in using BO for experiment design. Chemistry experiments are often done in large batches, and

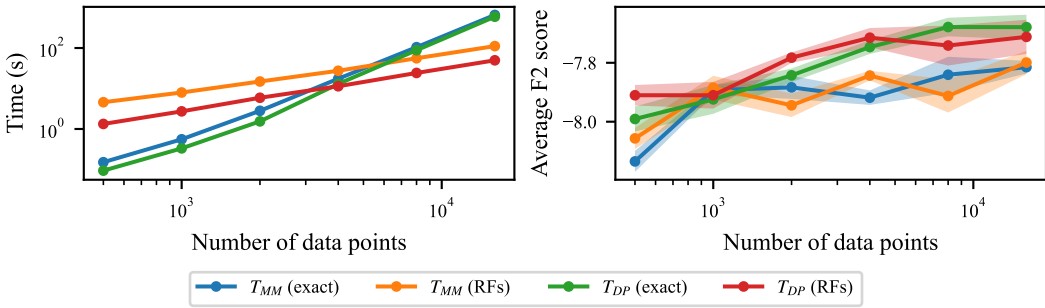

Figure 4: Run time and docking scores of BO using exact and approximate Thompson sampling. Solid lines are means over 5 trials, shaded regions are standard errors.

therefore algorithms which use functions sampled from a probabilistic model are of particular interest (Hernández-Lobato et al., 2017). To sample from a normal distribution $\mathcal{N}(\mu, K)$ one typically transforms i.i.d. samples $Z \sim \mathcal{N}(0, 1)$ via $\mu + K^{1/2}Z$. This requires computing $K^{1/2}$, and thereby causes exact GP sampling to scale cubically in the number of *evaluation* points. By approximating $K \approx \Phi^T \Phi$, our random features allow for approximate sampling in *linear* time. In this section we apply this to a real-world dataset.

As a demonstration, we consider a single round of selecting 100 molecules from a random sub-sample of $n$ molecules using *Thompson sampling*, a procedure for Bayesian optimization which selects molecules that maximize a function sampled from a GP prior using the $T_{MM}$ and $T_{DP}$ kernels. Similar to the setup from section 6.2, we use molecules and labels from the DOCKSTRING dataset. Molecules are represented as count fingerprints. $M = 5000$ random features used, with Rademacher $\Xi$ for $T_{MM}$ and no bias correction for $T_{DP}$. All other implementation details are the same as in the previous subsection. Figure 4 (left) shows that, as expected, exact Thompson sampling scales worse than approximate Thompson sampling with random features. Figure 4 (right) shows that using approximate instead of exact Thompson sampling does not seem to change the average F2 docking scores of the molecules chosen. This suggests that approximate Thompson sampling could fruitfully be applied to large datasets of molecules in Bayesian optimization tasks.

## 7 Discussion and conclusion

In this paper we presented two kinds of random features to estimate Tanimoto kernel matrices: one based on random hashes and another based on a power series expansion. To our knowledge, this is the first investigation into random features for the Tanimoto kernel. We theoretically analyze their approximation quality and demonstrate that they can effectively approximate the Tanimoto kernel matrices on realistic molecular fingerprint data. In the process we discovered a new Tanimoto-like kernel over all of $\mathbb{R}^d$ which is a promising substitute for the more established $T_{MM}$ on regression and optimization tasks.

Despite promising theoretical and experimental results, our random features do have some limitations. We found that it was difficult to efficiently vectorize the computation of the random features for $T_{MM}$, making them undesirably slow to compute. For $T_{DP}$, we were able to exhibit an error bound on the spectral norm which depends on certain choices for the base sketch and sketch dimensions; however, it is unclear whether these choices are optimal in practice. Nonetheless, in Appendix D.5 we prove that *exact* low-rank factorizations of $T_{MM}$ and $T_{DP}$ kernel matrices are not possible; this means that follow-up works could reduce but never eliminate the approximation error.

We are most optimistic about the potential of our random features to be applied in Bayesian optimization, in particular by enabling scalable approximate sampling from GP posteriors (Wilson et al., 2020). Although we briefly explored this technique in section 6.3, in the future it could allow for sample-efficient Bayesian algorithms for complex tasks like Pareto frontier exploration and diverse optimization using the Bayesian algorithm execution framework (Neiswanger et al., 2021). These tasks are highly relevant to real-world drug discovery and there are scant new methods poised to solve them in a sample-efficient way. We hope that the methods presented in this paper enable impactful, large-scale applications of the Tanimoto kernel and its two extensions in chemoinformatics.

## Acknowledgments and Disclosure of Funding

We thank Isaac Reid and Zhen Ning David Liu for helpful discussions. Austin Tripp acknowledges funding via a C T Taylor Cambridge International Scholarship and the Canadian Centennial Scholarship Fund. Sukriti Singh acknowledges funding from the UK Engineering and Physical Sciences Research Council. José Miguel Hernández-Lobato acknowledges support from a Turing AI Fellowship under grant EP/V023756/1.

*Author contributions:* the initial idea of using random hashes to produce random features for the Tanimoto kernel came from Sergio. Austin came across Ioffe (2010) and realized his proposed hash could be used to extend the scheme to $T_{MM}$ for general non-negative vectors. Austin derived and proved the variance statement from Theorem 3.1. Sergio proposed and proved Theorem 4.1 and Corollary 4.2 for the $T_{DP}$ kernel. Austin proposed and proved Proposition 4.3 after reading Charikar (2002). Sergio developed the random features for the prefactor (section 4.1) and the overall schema for the random features (section 4.2). Sergio proposed and proved Theorem 4.7. All experiments were designed and performed by Austin. Sukriti helped with some experiments which ultimately did not appear in this version of the manuscript. Sergio and José Miguel provided advising throughout the project. Writing was done jointly, but mostly by Austin and Sergio.

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

## Outline of appendices

- Appendix A comments on the items in the NeurIPS Paper Checklist.

- Appendix B summarizes the notation used in this paper.

- Appendix C explains TREESKETCH from Ahle et al. (2020), which is used in the proof of Theorem 4.7.

- Appendix D provides proofs for all theoretical results in the paper.

- Appendix E explains possible techniques to correct the bias of the $T_{DP}$ kernel which are mentioned briefly in section 4.

- Appendix F gives additional details about the experiments from section 6 and presents some supplementary results.

# A Paper Checklist

Here we explicitly comment on all areas of the NeurIPS paper checklist.

**Claims**    The key claims in this paper are the creation of random features for the Tanimoto kernel and the creation of a new continuous kernel. The claims are mainly supported theoretically with proofs, but we also test our random features experimentally and show that they are usable with real-world data.

**Code of Ethics + broader impacts**    Read and acknowledged. Our work is fairly abstract and theoretical and we do not work with human-related data, so we do not foresee significant direct ethical impacts (positive or negative).

**Limitations**    Our work is primarily theoretical, so the main strength (random features with low approximation error) is also the main weakness (the approximation error could plausibly be lower). Random features are well-studied as a general approach so the strengths and weaknesses are generally well-understood (Liu et al., 2021). For this reason we did not include an explicit limitations section. However, we did mention some other limitations in section 7.

**Theory**    Our key theorems have minimal assumptions which are clearly stated. All theorems are proved in Appendix D.

**Experiments**    Code to reproduce the experiments is available at: https://github.com/AustinT/tanimoto-random-features-neurips23.

**Training details**    The experiments in this paper have minimal details and the important details are specified in Appendix F. All unspecified details should be easy to find in the code (https://github.com/AustinT/tanimoto-random-features-neurips23).

**Error bars**    Our tables and figures include error bars.

**Compute**    The compute costs of the experiments in this paper were quite modest and were run on a single machine with no GPU usage. The approximate computational times were:

- Section 6.1: a script to produce all the plots took approximately 24 h to run. The long runtime was mostly due to computing errors for many settings of $R$ and $p$.
- Section 6.2: for $T_{MM}$, SVGP and each RFGP inference took approximately 1 h (total 3 h per target). For $T_{DP}$, SVGP took 0.5 h while RFGP took 0.3 h (total 1.5 h per target). Multiplying this by 5 targets and 5 trials gives a total compute time of $\approx$112 h.
- Section 6.3: this experiment was quite fast: total runtime was perhaps 5 h.

**Reproducibility**    Our contribution is reproducible via the statements of our random features and our code. We aim for a high standard of reproducibility: the exact commands to reproduce the results are stated explicitly in our code's README file, and the raw data behind all plots in this paper is included alongside the code.

**Safeguards**    We believe there is nothing high-risk which we need to safeguard.

**Licenses**    We cite the assets which we use in the paper.

**Assets**    We are not releasing assets.

**Humans Subjects / IRB Approvals**    Not applicable to our paper.

# B  Notation

We use $\|x\|$ to denote the Euclidean norm of a vector $x$, $\|A\|_{\mathrm{op}}$ for the spectral norm and $\|A\|_F$ for the Frobenius norm of a matrix $A$. $x_i$ denotes the $i$th element of a vector $x$, while $x^{(i)}$ denotes the $i$th vector in a list of vectors. We denote $\mathrm{nnz}(\cdot)$ the number of non-zero entries in a vector or matrix. For two functions $f, g$ we say $f(z) = O(g(z))$ if there is a constant $C$, such that $0 \leq f(z) \leq Cg(z)$ for $z$ large enough. Similarly, $f(z) = \Omega(g(z))$ if there is a constant $C$, such that $f(z) \geq Cg(z)$ for $z$ large enough. $\tilde{O}$ and $\tilde{\Omega}$ omit poly-logarithmic factors.

Throughout the paper, we use $d$ for the dimension of input vectors, $n$ for the number of samples in the dataset, and $M$ for the dimension of the sketch or random features.

## C Definition of TREESKETCH

For simplicity, we shall define TREESKETCH (Ahle et al., 2020) for inputs which are $r$-fold tensor products $x^{(1)} \otimes \cdots \otimes x^{(r)}$ where $r$ is a power of two, and each $x^{(i)} \in \mathbb{R}^d$ is of the same dimension $d$.

The main ingredients will be two *base* sketches, one for the leaves of the tree, and one for internal nodes. The leaf sketch, $T_{\text{base}}$ is a random matrix in $\mathbb{R}^{m \times d}$. The internal sketch, $S_{\text{base}}$, is a random matrix in $\mathbb{R}^{m \times m^2}$, which can be rapidly applied to the tensor product of two vectors in $\mathbb{R}^m$. It is possible to instantiate TreeSketch using simple sketches such as COUNTSKETCH, for the leaves, and TENSORSKETCH for internal nodes. However, our theory we assumes that $T_{\text{base}}$ is OSNAP (Nelson and Nguyên, 2013), and $S_{\text{base}}$ is TENSORSHRT, both of which enjoy a useful spectral property.

Having picked the base sketches, we can define for any power of two $q \geq 2$, a random map $Q^q : \mathbb{R}^{m^q} \to \mathbb{R}^m$ whose action on a tensor product $v^{(1)} \otimes \cdots \otimes v^{(q)}$ with $v^{(i)} \in \mathbb{R}^m$ is given by the following recursion:

$$Q^q = Q^{q/2}(S_1^q(v^{(1)} \otimes v^{(2)}) \otimes S_2^q(v^{(3)} \otimes v^{(4)}) \otimes \cdots \otimes S_{q/2}^q(v^{(q-1)} \otimes v^{(q)})).$$

Here, the matrices $(S_j^i)$ are independent copies of $S_{\text{base}}$. The action of TREESKETCH $\Pi^r$ on $x^{(1)} \otimes \cdots \otimes x^{(r)}$ is then defined by

$$\Pi^r(x^{(1)} \otimes \cdots \otimes x^{(r)}) = Q^r(T_1(x^{(1)}) \otimes \cdots \otimes T_r(x^{(r)}))$$

where the matrices $(T_i)$ are independent copies of $T_{\text{base}}$. Figure C.1 gives a schematic view of the computational tree for a tensor product with $r = 4$.

Whilst the action of $\Pi^r$ was only defined for tensor products, this can be extended to arbitrary vectors in $\mathbb{R}^{d^q}$. Ahle et al. (2020) prove that the resulting linear sketch has many desirable subspace embedding properties, some of which are used in the proof of Theorem 4.7.

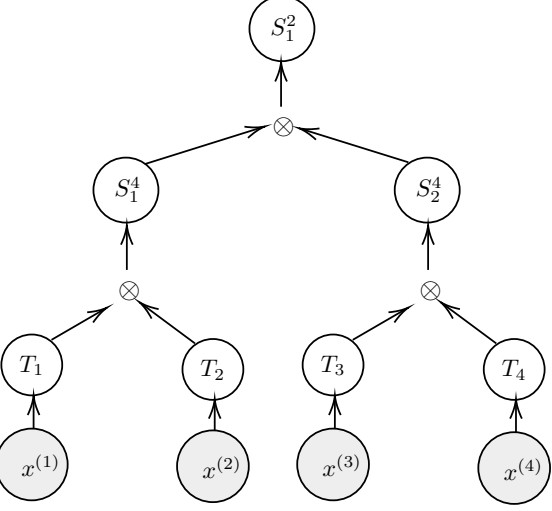

Figure C.1: Schematic view of the computational tree for TREESKETCH for a tensor product of four vectors $x^{(1)} \otimes \cdots \otimes x^{(4)}$.

## D  Proofs

### D.1  Proof of Theorem 3.1

For simplicity, we drop the subscripts $h, \Xi$ from probabilities and expectations. To show that the random features are unbiased, we re-write the expectation:

$$
\begin{aligned}
&\mathbb{E}\left[\phi_{\Xi,h}(x) \cdot \phi_{\Xi,h}(x')\right] \\
&= \mathbb{P}(h(x) = h(x'))\mathbb{E}\left[\Xi_{h(x)}\Xi_{h(x')}|h(x) = h(x')\right] \\
&\quad + \mathbb{P}(h(x) \neq h(x'))\mathbb{E}\left[\Xi_{h(x)}\Xi_{h(x')}|h(x) \neq h(x')\right] \\
&= \mathbb{P}(h(x) = h(x'))\mathbb{E}\left[\Xi_{h(x)}^2\right] & (\Xi_{h(x)} = \Xi_{h(x')}) \\
&\quad + \mathbb{P}(h(x) \neq h(x'))\mathbb{E}\left[\Xi_{h(x)}\right]\mathbb{E}\left[\Xi_{h(x')}\right] & (\Xi_{h(x)}, \Xi_{h(x')} \text{ independent}) \\
&= 1 \cdot \mathbb{P}(h(x) = h(x')) + 0 \cdot \mathbb{P}(h(x) \neq h(x')) & (\text{assumed moments of } \xi) \\
&= T_{MM}(x, x') & (h(x) \text{ is an unbiased hash for } T_{MM})
\end{aligned}
$$

Because in general $\mathbb{V}[X] = \mathbb{E}[X^2] - \mathbb{E}[X]^2$, and we have $\mathbb{E}\left[\phi_{\Xi,h}(x) \cdot \phi_{\Xi,h}(x')\right] = T_{MM}(x, x')$, we only need to compute $\mathbb{E}\left[\left(\phi_{\Xi,h}(x) \cdot \phi_{\Xi,h}(x')\right)^2\right]$. This can be done using a similar decomposition:

$$
\begin{aligned}
\mathbb{E}\left[\left(\phi_{\Xi,h}(x) \cdot \phi_{\Xi,h}(x')\right)^2\right] &= \mathbb{P}(h(x) = h(x'))\mathbb{E}\left[\left(\phi_{\Xi,h}(x) \cdot \phi_{\Xi,h}(x')\right)^2 |h(x) = h(x')\right] \\
&\quad + \mathbb{P}(h(x) \neq h(x'))\mathbb{E}\left[\left(\phi_{\Xi,h}(x) \cdot \phi_{\Xi,h}(x')\right)^2 |h(x) \neq h(x')\right] \\
&= T_{MM}(x, x')\mathbb{E}\left[\Xi_{h(x)}^4\right] \\
&\quad + (1 - T_{MM}(x, x'))\mathbb{E}\left[\Xi_{h(x)}^2\right]\mathbb{E}\left[\Xi_{h(x')}^2\right] \\
&= T_{MM}(x, x')\mathbb{E}\left[\Xi_{h(x)}^4\right] + (1 - T_{MM}(x, x'))
\end{aligned}
$$

Here, the last step follows from the moments of $\Xi$ which were fixed by assumption. Subtracting $T_{MM}(x, x')^2$ from the above yields the expression for the variance.

By Jensen's inequality, $\mathbb{E}[\xi^4] \geq \mathbb{E}[\xi^2]^2$, and $\mathbb{E}[\xi^2] = 1$ by assumption, so $\mathbb{E}[\xi^4] \geq 1$. Substituting $\mathbb{E}[\xi^4] = 1$ into the equation for $\mathbb{V}(\phi_{\Xi,h}(x) \cdot \phi_{\Xi,h}(x'))$ gives the lower bound and completes the proof.

### D.2  Proof of Theorem 4.1 and Corollary 4.2

Take any pair of inputs $x, y \neq 0$ in $\mathbb{R}^d$. By the Cauchy–Schwarz inequality, we have

$$
\left|\frac{x \cdot y}{\|x\|^2 + \|y\|^2}\right| \leq \frac{\|x\|\|y\|}{\|x\|^2 + \|y\|^2} \leq \left(\|x\|/\|y\| + \frac{1}{\|x\|/\|y\|}\right)^{-1} \leq \frac{1}{2}. \tag{15}
$$

Now, we can write the function $T_{DP}$ as

$$
T_{DP}(x, y) = \frac{x \cdot y}{\|x\|^2 + \|y\|^2 - x \cdot y} = \frac{\frac{x \cdot y}{\|x\|^2 + \|y\|^2}}{1 - \frac{x \cdot y}{\|x\|^2 + \|y\|^2}} \tag{16}
$$

$$
= \frac{x \cdot y}{\|x\|^2 + \|y\|^2}\left(\frac{1}{1 - \frac{x \cdot y}{\|x\|^2 + \|y\|^2}}\right) \tag{17}
$$

$$
= \frac{x \cdot y}{\|x\|^2 + \|y\|^2}\sum_{r=0}^{\infty}\left(\frac{x \cdot y}{\|x\|^2 + \|y\|^2}\right)^r \tag{18}
$$

$$
= \sum_{r=1}^{\infty}\left(\frac{x \cdot y}{\|x\|^2 + \|y\|^2}\right)^r, \tag{19}
$$

where in the identity (18) we use the bound in (15) to assert that the series is bounded by $2^{-r}$ and thus absolutely convergent.

Furthermore, because $(x, y) \mapsto x \cdot y$ is a positive definite kernel, and so is

$$(x, y) \mapsto \frac{1}{\|x\|^2 + \|y\|^2} = \int_0^\infty e^{-(\|x\|^2 + \|y\|^2)t} dt,$$

each summand in the power series is positive definite because it is a product of positive definite kernels. Hence, as sums and limits of positive definite kernels are positive definite, and the power series is convergent, $T_{DP}$ is positive definite in the space $\mathbb{R}^d \setminus \{0\}$. The extension to include the vector $0 \in \mathbb{R}^d$ is straightforward, as any Gram matrix including this vector is block diagonal. We conclude that $T_{DP}$ is a positive definite kernel in $\mathbb{R}^d$.

To prove Corollary 4.2, note that by the Moore–Aronszajn theorem, there exists an RKHS of functions $(\mathcal{H}, \langle \cdot \rangle_{\mathcal{H}})$ on $\mathbb{R}^d$ with reproducing kernel $T_{DP}$, such that for any $x, y \in \mathbb{R}^d$, $\langle T_{DP}(x, \cdot), T_{DP}(y, \cdot) \rangle_{\mathcal{H}} = T_{DP}(x, y)$. Then, observe that

$$
\begin{aligned}
\|T_{DP}(x, \cdot) - T_{DP}(y, \cdot)\|_{\mathcal{H}}^2 &= \langle T_{DP}(x, \cdot) - T_{DP}(y, \cdot), T_{DP}(x, \cdot) - T_{DP}(y, \cdot) \rangle_{\mathcal{H}} \\
&= \langle T_{DP}(x, \cdot), T_{DP}(x, \cdot) \rangle_{\mathcal{H}} + \langle T_{DP}(y, \cdot), T_{DP}(y, \cdot) \rangle_{\mathcal{H}} \\
&\quad - 2 \langle T_{DP}(x, \cdot), T_{DP}(y, \cdot) \rangle_{\mathcal{H}} \\
&= 2 - 2 T_{DP}(x, y),
\end{aligned}
$$

where in the final identity, we use the fact that $T_{DP}(x, x) = 1$ for all $x \in \mathbb{R}^d$. Dividing by 2 and taking square roots on both sides, we obtain

$$\left\| \frac{1}{2} T_{DP}(x, \cdot) - \frac{1}{2} T_{DP}(y, \cdot) \right\|_{\mathcal{H}} = \sqrt{1 - T_{DP}(x, y)},$$

where the RKHS norm on the left is clearly a distance metric.

### D.3 Proofs and derivations for prefactor random features from section 4.1

#### D.3.1 Scalar random features (Lemma 4.4)

First, we present a derivation for the scalar random features for the prefactor $(\|x\|^2 + \|x'\|^2)^{-r}$ from Lemma 4.4. As the kernel is a function of $\|x\|^2$ and $\|x'\|^2$, we can deal, without loss of generality, with the one-dimensional case. We begin by observing that for any $a, b > 0$,

$$
\begin{aligned}
\left( \frac{1}{a+b} \right)^r &= \int_0^\infty e^{(1/2-a)t} e^{(1/2-b)t} \frac{t^{r-1} e^{-t}}{\Gamma(r)} dt \\
&= \int_0^\infty e^{(1/2-a)t} e^{(1/2-b)t} c^{-r} e^{(c-1)t} \frac{c^r t^{r-1} e^{-ct}}{\Gamma(r)} dt \\
&= \int_0^\infty e^{(1/2-a)t} e^{(1/2-b)t} \frac{c^{-s} e^{(c-1)t} t^{r-s} \Gamma(s)}{\Gamma(r)} \frac{c^s t^{s-1} e^{-ct}}{\Gamma(s)} dt.
\end{aligned}
\tag{20}
$$

Defining the function

$$\varphi_Z(a) = e^{(1/2-a)Z} Z^{(r-s)/2} \left( \frac{c^{-s} e^{(c-1)Z} \Gamma(s)}{\Gamma(r)} \right)^{1/2}$$

and letting $Z \sim \text{Gamma}(s, c)$, we recognise the right hand side of (20) as the expectation of $\varphi_Z(a) \varphi_Z(b)$. Hence,

$$\left( \frac{1}{a+b} \right)^r = \mathbb{E}(\varphi_Z(a) \varphi_Z(b)).$$

*This proves Lemma 4.4.*

#### D.3.2 QMC random features (Lemma 4.5)

Next we motivate and derive our QMC random features for the prefactor, proving Lemma 4.5. It is possible to sketch $a$ with a vector $\varphi(a) = \frac{1}{\sqrt{M}}(\varphi_{Z_1}(a), \ldots, \varphi_{Z_M}(a))$ where $Z_1, \ldots, Z_M$ are independent copies of $Z$. This makes the kernel approximation $\varphi(a) \cdot \varphi(b)$ a Monte Carlo estimator

of the expectation, with error decreasing with the dimension $M$ of the sketch at the standard rate $O(M^{-1/2})$.

However, we shall instead consider a Quasi-Monte Carlo (QMC) estimator with an error decreasing at the faster rate $O(M^{-1})$. Let $\gamma_{s,c}$ be the inverse cumulative distribution function of a Gamma$(s, c)$ random variable. With a change of variables, we can write the integral (20) as

$$\left(\frac{1}{a+b}\right)^r = \frac{c^{-s}\Gamma(s)}{\Gamma(r)} \int_0^1 e^{-(a+b-c)\gamma_{s,c}(u)}(\gamma_{s,c}(u))^{r-s}du. \tag{21}$$

Fix $n > 0$, $u \in [0, 1]$, and let $u_i = (u + i/M) - \lfloor u + i/M \rfloor$ for $i = 1, \dots, M$. We can now define features $\phi_{u,r}(a) \in \mathbb{R}^M$, as in Section 4.2:

$$\phi_{u,r,i}(a) = \frac{1}{\sqrt{M}}\sqrt{\frac{c^{-s}\Gamma(s)}{\Gamma(r)}}e^{-(a-c/2)\gamma_{s,c}(u_i)}(\gamma_{s,c}(u_i))^{(r-s)/2}.$$

A QMC estimator for $(a + b)^{-r}$ is given by the dot product $\phi_u(a)^T\phi_u(b)$. This estimator is unbiased:

**Lemma.** *If $u$ is random and $\mathcal{U}(0, 1)$, then for all $a, b > 0$,*

$$\left(\frac{1}{a+b}\right)^r = \mathbb{E}(\phi_{u,r}(a) \cdot \phi_{u,r}(b)).$$

*Proof.* By linearity of expectation,

$$\mathbb{E}(\phi_{u,r}(a) \cdot \phi_{u,r}(b)) = \frac{1}{M}\sum_{i=1}^M \mathbb{E}\left[\frac{c^{-s}\Gamma(s)}{\Gamma(r)}e^{-(a+b-c)\gamma_{s,c}(u_i)}(\gamma_{s,c}(u_i))^{r-s}\right].$$

Observing that each $u_i \sim \mathcal{U}(0, 1)$, the result follows by Eq. (21). $\qquad\square$

This lemma is equivalent to Lemma 4.5, thereby proving it.

### D.3.3   Relative error bound (Lemma 4.6)

In the following lemma, we establish a basic Quasi-Monte Carlo error bound.

**Lemma D.1.** *For any $u \in [0, 1]$, $c \le (a + b)$, $0 < s \le r$,*

$$\left|\frac{1}{(a+b)^r} - \phi_{u,r}(a) \cdot \phi_{u,r}(b)\right| \le \frac{2}{M}\frac{c^{-s}\Gamma(s)}{\Gamma(r)}\left(\frac{r-s}{e(a+b-c)}\right)^{r-s}.$$

*Proof.* Consider the QMC approximation of an integral $\int_0^1 f(x)dx$ for a function $f$ of bounded variation, using a regular net of $n$ points. It is well known that the error of this approximation is bounded above by $V(f)/n$ where $V$ is the total variation norm. Hence,

$$\left|\frac{1}{(a+b)^r} - \phi_{u,r}(a) \cdot \phi_{u,r}(b)\right| \le \frac{1}{n}\frac{c^{-s}\Gamma(s)}{\Gamma(r)}V(f)$$

for $f(u) = e^{-(a+b-c)\gamma_{s,c}(u)}(\gamma_{s,c}(u))^{r-s}$ on $0 \le u \le 1$. As $\gamma_{s,c}$ is monotone increasing, $f$ is unimodal tending to 0 as $u \to 0$ and $u \to 1$. Thus $V(f) = 2\max_{0\le u\le1} f(u)$. The maximum of $f$ is attained where $\gamma_{s,c}(u) = (r - s)/(a + b - c)$, hence

$$V(f) = 2\left(\frac{r-s}{e(a+b-c)}\right)^{r-s}. \qquad\square$$

We are interested in sketching $(\|x^{(i)}\|^2 + \|x^{(j)}\|^2)^{-r}$ for every pair of inputs $x^{(i)}, x^{(j)}$ in a dataset $\{x^{(1)}, \dots, x^{(n)}\}$. The previous lemma can be used to choose values $c$ and $s \le r$ which minimise the relative error

$$E_{i,j} = \frac{\phi_{u,r}(x^{(i)}) \cdot \phi_{u,r}(x^{(j)}) - (\|x^{(i)}\|^2 + \|x^{(j)}\|^2)^{-r}}{(\|x^{(i)}\|^2 + \|x^{(j)}\|^2)^{-r}}.$$

The kernel $k(x, y)$ is invariant to scaling $x$ and $y$, *i.e.*, $k(x, y) = k(\ell x, \ell y)$ for all $\ell > 0$. Thus, if we are interested in approximating the Gram matrix for $\{x^{(1)}, \ldots, x^{(n)}\}$, we may assume without loss of generality that $\max_i \|x^{(i)}\|^2 = 1$ and let $\zeta = \min_i \|x^{(i)}\|^2$. Here, $\zeta$ will act as a condition number for the dataset, which will determine the error of the random feature approximation.

By Lemma D.1, we have

$$
\begin{aligned}
|E_{i,j}| &\leq (\|x^{(i)}\|^2 + \|x^{(j)}\|^2)^r \frac{2}{M} \frac{c^{-s}\Gamma(s)}{\Gamma(r)} \left( \frac{r-s}{e(\|x^{(i)}\|^2 + \|x^{(j)}\|^2 - c)} \right)^{r-s} \\
&= (\|x^{(i)}\|^2 + \|x^{(j)}\|^2)^r \frac{2}{M} \frac{c^{-s}\Gamma(s)(r-s)^{r-s}}{\Gamma(r)e^{(r-s)}} \left( \frac{1}{\|x^{(i)}\|^2 + \|x^{(j)}\|^2 - c} \right)^{r-s} \\
&= \frac{2}{M} \frac{\Gamma(s)(r-s)^{r-s}}{\Gamma(r)e^{(r-s)}} \left( 1 - \frac{c}{\|x^{(i)}\|^2 + \|x^{(j)}\|^2} \right)^{s-r} \left( \frac{c}{\|x^{(i)}\|^2 + \|x^{(j)}\|^2} \right)^{-s}.
\end{aligned}
$$

The last two factors are log-convex in $c/(\|x^{(i)}\|^2 + \|x^{(j)}\|^2) > 0$. Hence, for a fixed $c$, the product

$$
\left( 1 - \frac{c}{\|x^{(i)}\|^2 + \|x^{(j)}\|^2} \right)^{s-r} \left( \frac{c}{\|x^{(i)}\|^2 + \|x^{(j)}\|^2} \right)^{-s}
$$

is maximised at either $\|x^{(i)}\|^2 + \|x^{(j)}\|^2 = 2$ or $\|x^{(i)}\|^2 + \|x^{(j)}\|^2 = 2\zeta$. Therefore,

$$
|E_{i,j}| \leq \frac{2}{M} \frac{\Gamma(s)(r-s)^{r-s}}{\Gamma(r)e^{(r-s)}} \left[ \left( 1 - \frac{c}{2} \right)^{s-r} \left( \frac{c}{2} \right)^{-s} \vee \left( 1 - \frac{c}{2\zeta} \right)^{s-r} \left( \frac{c}{2\zeta} \right)^{-s} \right].
$$

Using again the log-convexity of $y \mapsto (1-y)^{s-r}y^{-s}$, we find that

$$
\left( 1 - \frac{c}{2} \right)^{s-r} \left( \frac{c}{2} \right)^{-s}
$$

is minimised as a function of $c$ at $c_1 = 2s/r$. And

$$
\left( 1 - \frac{c}{2\zeta} \right)^{s-r} \left( \frac{c}{2\zeta} \right)^{-s}
$$

is minimised at $c_2 = 2\zeta s/r$. Recall that Lemma D.1 requires $0 \leq c \leq \|x^{(i)}\|^2 + \|x^{(j)}\|^2$ for all $i, j$ or $0 \leq c \leq 2\zeta$, which is satisfied by $c_2$ when $s \leq r$, but not necessarily by $c_1$. Thus, it is reasonable to set $c = c_2$, which leads to

$$
\begin{aligned}
|E_{i,j}| &\leq \frac{2}{M} \frac{\Gamma(s)(r-s)^{r-s}}{\Gamma(r)e^{(r-s)}} \left[ \left( 1 - \frac{\zeta s}{r} \right)^{s-r} \left( \frac{\zeta s}{r} \right)^{-s} \vee \left( 1 - \frac{s}{r} \right)^{s-r} \left( \frac{s}{r} \right)^{-s} \right] \\
&\leq \frac{2}{M} \frac{\Gamma(s)(r-s)^{r-s}r^r s^{-s}}{\Gamma(r)e^{(r-s)}} \left[ (r - \zeta s)^{s-r} \zeta^{-s} \vee (r-s)^{s-r} \right] \\
&\leq \frac{2}{M} \frac{\Gamma(s)(s/e)^{-s}}{\Gamma(r)(r/e)^{-r}} \left[ \left( \frac{r - \zeta s}{r - s} \right)^{s-r} \zeta^{-s} \vee 1 \right].
\end{aligned}
$$

When $\zeta = 1$ (all inputs $x^{(i)}$ have the same norm), setting $s = r$ leads to an error bound of $4/n$. However, small values of $\zeta$ can make the upper bound blow up unless we tune $s$ suitably. For a given value of $\zeta$, this upper bound could be maximised numerically over $0 < s \leq r$. However, setting $s = r\zeta$, we obtain

$$
\begin{aligned}
|E_{i,j}| &\leq \frac{2}{M} \frac{\Gamma(r\zeta)(r\zeta/e)^{-r\zeta}}{\Gamma(r)(r/e)^{-r}} \left[ \left( \frac{1-\zeta^2}{1-\zeta} \right)^{r(\zeta-1)} \zeta^{-r\zeta} \vee 1 \right] \\
&\leq \frac{2}{M} \frac{\Gamma(r\zeta)\zeta^{-r\zeta}}{\Gamma(r)} (r/e)^{r(\zeta-1)} \left[ \left( \left( \frac{1-\zeta^2}{1-\zeta} \right)^{\zeta-1} \zeta^{-\zeta} \right)^r \vee 1 \right] \\
&\leq \frac{2}{M} \frac{\Gamma(r\zeta)\zeta^{-r\zeta}}{\Gamma(r)} (r/e)^{r(\zeta-1)} \rho^r.
\end{aligned}
$$

where $\rho \approx 1.2$ is defined by

$$\rho := \max_{0 \leq \zeta \leq 1} \left( \frac{1 - \zeta^2}{1 - \zeta} \right)^{\zeta - 1} \zeta^{-\zeta}.$$

This proves the first inequality in Lemma 4.6.

To conclude the proof of Lemma 4.6, we study the behaviour of the upper bound as it diverges when $\zeta \to 0$. Using the asymptotic $\Gamma(x) \sim 1/x$ as $x \to 0$, we have

$$\frac{2}{M} \frac{\Gamma(r\zeta)\zeta^{-r\zeta}}{\Gamma(r)} (r/e)^{r(\zeta-1)} \rho^r \sim (M\zeta)^{-1} \frac{2(r/e)^{-r}\rho^r}{r\Gamma(r)}$$

where $\frac{2(r/e)^{-r}\rho^r}{r\Gamma(r)}$ is bounded for $r \geq 1$. Therefore, the upper bound is $O(M^{-1}\zeta^{-1})$. This completes the proof.

### D.4 Proof of Theorem 4.7

We begin by defining the stable rank, a common notion of effective dimension for a data matrix $A \in \mathbb{R}^{d \times n}$ or its associated Gram matrix $A^T A$.

**Definition D.2.** The *stable rank* $\mathrm{sr}(A)$ of a data matrix $A \in \mathbb{R}^{d \times n}$ is given by

$$\mathrm{sr}(A) = \frac{\|A\|_F^2}{\|A\|_{\mathrm{op}}^2}.$$

For a positive semi-definite matrix $K \in \mathbb{R}^{n \times n}$, define $\tilde{\mathrm{sr}}(K) = \mathrm{Tr}(K)/\|K\|_{\mathrm{op}}$.

The stable rank of $A$ is upper bounded by the rank, but it is insensitive to small singular values, so it can be much smaller than $\min(n, d)$. Note that $\mathrm{sr}(A) = \tilde{\mathrm{sr}}(A^T A)$. The function $\tilde{\mathrm{sr}}(K)$ extends the concept of stable rank to a general kernel matrix $K$.

We now re-state theorem 4.7 for convenience, and provide a proof:

**Theorem.** *For any $n \geq 1$, let $x^{(1)}, \ldots, x^{(n)} \in \mathbb{R}^d$ be a set of inputs with $\frac{\min_i \|x^{(i)}\|^2}{\max_i \|x^{(i)}\|^2} \geq \zeta$. Let $K$ be the matrix with entries $K_{i,j} = T_{DP}(x^{(i)}, x^{(j)})$. For all $\varepsilon > 0$, there exists an oblivious sketch $\Phi : \mathbb{R}^d \to \mathbb{R}^m$ with $m = \tilde{\Omega}(\tilde{sr}(K)/\varepsilon^2)$, such that*

$$\mathbb{P}_\Phi\left(\|\widehat{K} - K\|_{op} \geq \varepsilon\|K\|_{op}\right) \leq \frac{1}{poly(n)} \tag{22}$$

*where $\widehat{K}_{i,j} = \Phi(x^{(i)}) \cdot \Phi(x^{(j)})$. Furthermore, the sketch can be computed in time $\tilde{O}(\tilde{sr}(K)n\varepsilon^{-2} + nnz(X)\varepsilon^{-2} + n\zeta^{-1}\varepsilon^{-3})$.*

We begin by specifying the sketch $\Phi$. Define

$$\Phi(x) = \oplus_{r=1}^R \Phi^r(x)$$

where $\Phi^r(x) = \Pi^{r+1}(\phi_{u,r}(x) \otimes x^{\otimes r})$, $\Pi^r$ is a TREESKETCH (see appendix C) and $\phi_{u,r}(x)$ is the random feature expansion for the prefactor defined in Lemma 4.5. TREESKETCH (Ahle et al., 2020), must be instantiated with the base sketches OSNAP (Nelson and Nguyên, 2013) at the leaves ($T_{\mathrm{base}}$), and TENSORSRHT (Ahle et al., 2020) at internal nodes ($S_{\mathrm{base}}$). The choices for $R$, and the dimensions of $\Pi^r$ and $\phi_{u,r}$ will be made explicit in the proof.

For each $r \in [R]$, define matrices $K^r$, $\tilde{K}^r$, and $\widehat{K}^r$ in $\mathbb{R}^{n \times n}$ by

$$K_{i,j}^r = \left( \frac{x^{(i)} \cdot x^{(j)}}{\|x^{(i)}\|^2 + \|x^{(j)}\|^2} \right)^r,$$
$$\tilde{K}_{i,j}^r = (x^{(i)\otimes r} \otimes \phi_{u,r}(x^{(i)})) \cdot (x^{(i)\otimes r} \otimes \phi_{u,r}(x^{(i)})),$$
$$\widehat{K}_{i,j}^r = \Phi_r(x^{(i)}) \cdot \Phi_r(x^{(j)}),$$

Then, by the triangle inequality, we have

$$\|\widehat{K} - K\|_{\text{op}} \le \sum_{r=1}^{R} \|\widehat{K}^r - \tilde{K}^r\|_{\text{op}} + \sum_{r=1}^{R} \|\tilde{K}^r - K^r\|_{\text{op}} + \left\|\sum_{r=1}^{R} K^r - K\right\|_{\text{op}}$$

$$\le \sum_{r=1}^{R} \|\widehat{K}^r - \tilde{K}^r\|_{\text{op}} + \sum_{r=1}^{R} \|\tilde{K}^r - K^r\|_{\text{op}} + \sum_{r=R+1}^{\infty} \|K^r\|_{\text{op}}. \qquad (23)$$

We shall find high-probability bounds for each of the terms on the right in turn. For the last term, we can choose $R = C \log(\tilde{\text{sr}}(K)\varepsilon^{-1})$ with $C$ large enough, and apply the following upper bound

$$\sum_{r=R+1}^{\infty} \|K^r\|_{\text{op}} \le \sum_{r=R+1}^{\infty} \|K^r\|_F$$

$$\le \sum_{r=R+1}^{\infty} n \left( \max_{i,j \in [n]} \frac{\|x^{(i)} \cdot x^{(j)}\|^r}{(\|x^{(i)}\|^2 + \|x^{(j)}\|^2)^r} \right)^{1/2}$$

$$\le \sum_{r=R+1}^{\infty} n 2^{-r/2}$$

$$\le n 2^{-(R-1)/2} \le \frac{n\varepsilon}{3\tilde{\text{sr}}(K)} \le \frac{\varepsilon\|K\|_{\text{op}}}{3}, \qquad (24)$$

where the final inequality follows from the fact that $\tilde{\text{sr}}(K) = n/\|K\|_{\text{op}}$, as the kernel matrix $K$ has ones on the diagonal.

The terms in the second sum of (23) may be written as

$$\|\tilde{K}^r - K^r\|_{\text{op}} = \|\tilde{E} \odot K^r\|_{\text{op}}$$

where $\odot$ denotes the Hadamard product and $E$ is the error matrix defined in (10). Noting that $E$ is symmetric, we can apply Corollary 11 in Ando et al. (1987) to assert that

$$\|\tilde{K}^r - K^r\|_{\text{op}} \le \max_i |E_{i,i}| \|K^r\|_{\text{op}}.$$

By the triangle inequality, for each $r \ge 1$, we have $\|K^r\|_{\text{op}} \le \|K\|_{\text{op}}$. Hence, with the choice $M = \Omega(R\zeta^{-1}\varepsilon^{-1})$ for the dimension of $\phi_{u,r}$, and applying Lemma 4.6 we obtain

$$\sum_{r=1}^{R} \|\tilde{K}^r - K^r\|_{\text{op}} \le \frac{\varepsilon\|K\|_{\text{op}}}{3}. \qquad (25)$$

Finally, we turn to the first term in (23). Let $A_r$ be a matrix with columns $x^{(i)\otimes r} \otimes \phi_{u,r}(x^{(i)})$ for $i = 1, \ldots, n$. We can rewrite the error in question

$$\|\widehat{K}^r - \tilde{K}^r\|_{\text{op}} = \|(\Pi^{r+1} A_r)^T \Pi^{r+1} A_r - A_r^T A_r\|_{\text{op}}$$

In Lemma D.3, we establish that $\|A_r\|_F^2 \le (1+\varepsilon)\text{Tr}(K)$ and $\|A_r\|_{\text{op}}^2 \le (1+\varepsilon)\|K\|_{\text{op}}$. Let $\delta = 1/\text{poly}(n)$ denote the error tolerance. Choose the dimension of the sketch $\Pi^{r+1}$ to be $m_r = \Omega\left(\frac{R^2 r^4 \tilde{\text{sr}}(K)}{\varepsilon^2} \log^3\left(\frac{n(d \vee M)}{\varepsilon\delta}\right)\right)$ and the sparsity parameter of OSNAP as $s = \frac{R^2 r^4}{\varepsilon^2} \log^3\left(\frac{n(d \vee M)}{\varepsilon\delta}\right)$. Then, Lemmata 32–34 in Ahle et al. (2020) ensure that,

$$\mathbb{P}\left(\|\widehat{K}^r - \tilde{K}^r\|_{\text{op}} \ge \frac{\varepsilon\|K\|_{\text{op}}}{3R}\right) = \mathbb{P}\left(\|(\Pi^{r+1} A_r)^T \Pi^{r+1} A_r - A_r^T A_r\|_{\text{op}} \ge \frac{\varepsilon\|K\|_{\text{op}}}{3R}\right) \le \frac{1}{\text{poly}(n)}.$$

Taking a union bound, we obtain

$$\mathbb{P}\left(\sum_{r=1}^{R} \|\widehat{K}^r - \tilde{K}^r\|_{\text{op}} \ge \frac{\varepsilon\|K\|_{\text{op}}}{3}\right) \le \frac{1}{\text{poly}(n)}. \qquad (26)$$

Finally, combining the bounds (26), (25), and (24), we obtain

$$\mathbb{P}\left(\|\widehat{K} - K\|_{\text{op}} \ge \varepsilon\|K\|_{\text{op}}\right) \le \frac{1}{\text{poly}(n)}.$$

**Dimension of the sketch** $\Phi$**:** The total dimension of the sketch is $m = m_1 + \cdots + m_R$, where $m_r = \Omega\left(\frac{R^2 r^4 \tilde{\mathrm{sr}}(K)}{\varepsilon^2} \log^3\left(\frac{n(d \vee M)}{\varepsilon \delta}\right)\right)$ with $R = C \log(\tilde{\mathrm{sr}}(K)\varepsilon^{-1})$. Hence, ignoring poly-logarithmic factors, the sketch has dimension $m = \tilde{\Omega}(\tilde{\mathrm{sr}}(K)/\varepsilon^2)$.

**Runtime:** We first estimate the runtime of applying $\Phi^r$. Applying OSNAP with sparsity parameter $s$ to a vector $w \in \mathbb{R}^d$ takes $O(s\mathrm{nnz}(w))$ time. Thus, applying $r$ independent OSNAP sketches to each $x^{(i)}$ for $i \in [n]$ takes $O(s r \mathrm{nnz}(X))$ time. Applying OSNAP to $\phi_{u,r}(x^{(i)})$ for $i \in [n]$ takes $O(nsM)$ time. Then, applying one TENSORSRHT sketch takes $O(m_r \log m_r)$ time, and therefore applying all the necessary copies of $S_{\mathrm{base}}$ requires $O(rm_r \log m_r)$. In total, the sketch $\Phi^a tior$ may be computed in $O(nrm_r \log m_r + s\mathrm{nnz}(X) + nsM)$. Adding these runtimes over $r \in [R]$ gives us a total runtime which is $\tilde{O}(\tilde{\mathrm{sr}}(K)n\varepsilon^{-2} + \mathrm{nnz}(X)\varepsilon^{-2} + n\zeta^{-1}\varepsilon^{-3})$.

**Lemma D.3.** *For each $r = 1, \ldots, R$, we have $\|A_r\|_{op}^2 \leq (1+\varepsilon)\|K\|_{op}$ and $\|A_r\|_F^2 \leq (1+\varepsilon)\mathrm{Tr}(K)$.*

*Proof.* Note that $A_r^T A_r = \tilde{K}^r$. Hence, we aim to show that $\|\tilde{K}^r\|_{\mathrm{op}} \leq (1+\varepsilon)\|K\|_{\mathrm{op}}$, and that $\mathrm{Tr}(\tilde{K}^r) \leq (1+\varepsilon)\mathrm{Tr}(K)$. By the triangle inequality,

$$\|\tilde{K}^r\|_{\mathrm{op}} \leq \|\tilde{K}^r\|_{\mathrm{op}} + \|\tilde{K}^r - K^r\|_{\mathrm{op}} \leq (1+\varepsilon)\|K^r\|_{\mathrm{op}} \leq (1+\varepsilon)\|K\|_{\mathrm{op}},$$

where the penultimate inequality was shown in the proof of Theorem 4.7, and the final inequality is due to the fact that $K^r \preceq K$.

Then, by linearity of the trace,

$$\mathrm{Tr}(\tilde{K}^r) = \mathrm{Tr}(K^r) + \mathrm{Tr}(\tilde{K}^r - K^r) = \mathrm{Tr}(K^r) + \mathrm{Tr}(E \odot K^r). \tag{27}$$

And applying once more Corollary 11 of Ando et al. (1987), we have

$$\mathrm{Tr}(E \odot K^r) = \sum_{i=1}^n \sigma_i(E \odot K^r) \leq \max_{i \in [n]} |E_{i,i}| \sum_{i=1}^n \sigma_i(K^r) \leq \varepsilon \sum_{i=1}^n \sigma_i(K^r),$$

where $\sigma_i(\cdot)$ denotes the $i$th singular value. Using once more that $K^r \preceq K$ and hence $\sigma_i(K^r) \leq \sigma_i(K)$ for all $i \in [n]$, we obtain,

$$\mathrm{Tr}(E \odot K^r) \leq \varepsilon \mathrm{Tr}(K).$$

Plugging this into (27) yields $\mathrm{Tr}(\tilde{K}^r) \leq (1+\varepsilon)\mathrm{Tr}(K)$. $\qquad \square$

## D.5 Rank of Tanimoto Kernels

**Lemma D.4.** *There does not exist an exact finite-dimensional feature map for either $T_{MM}$ or $T_{DP}$.*

*Proof.* We will use a proof by contradiction, first focusing on $T_{MM}$. The setup for the contradiction is as follows: suppose there existed an exact feature map $f : \mathbb{R}^d \mapsto \mathbb{R}^M$ such that $T_{MM}(x,y) = f(x) \cdot f(y) \ \forall x, y \in \mathbb{R}^d$. This would imply that any kernel matrix between $n$ points $X$ could be written as an inner product

$$T_{MM}(X, X) = f(X)^T f(X).$$

Because $f$ outputs $M$ dimensional vectors this would imply the resulting matrix has rank at most $M$. Therefore, under this hypothesis it should not be possible to form a kernel matrix of more than $M$ inputs which is full-rank (invertible). Our contradiction will be to construct such a matrix.

We now present a way to construct a full-rank $T_{MM}$ kernel matrix with any number of points $n$. Consider the set of points $\{a^{(i)}\}_{i=1}^n$ where $a^{(i)} = (2n)^i \in \mathbb{R}$. For any $i$, we have that $T_{MM}(a^{(i)}, a^{(i)}) = 1$ and if $n \geq 2$ then

$$\sum_{j \neq i} |T_{MM}(a^{(i)}, a^{(j)})| = \sum_{j \neq i} (2n)^{-|i-j|} \leq \sum_{j \neq i} \frac{1}{2n} \leq \frac{1}{2},$$

meaning the kernel matrix is *strictly diagonally dominant* and therefore non-singular. Since such a construction exists for any $n$, setting $n = M + 1$ contradicts the assumption of an $M$-dimensional

feature map, and repeating this argument for all finite $M$ proves that no finite-dimensional feature map can exists for $T_{MM}$.

The proof for $T_{DP}$ is almost identical. Consider the same sequence of points as in the preceding paragraph. For any $i$, we have that $T_{DP}(a^{(i)}, a^{(i)}) = 1$ and

$$\sum_{j \neq i} |T_{DP}(a^{(i)}, a^{(j)})| = \sum_{j \neq i} \frac{1}{(2n)^{|i-j|} + (2n)^{-|i-j|} - 1} \leq \sum_{j \neq i} \frac{1}{2n - 1} < 1 \,,$$

meaning the kernel matrix for $T_{DP}$ is also strictly diagonally dominant, thereby also precluding the existence of an $M$-dimensional feature map for finite $M$.

$\square$

This result suggests that the best we can hope for is an *approximate* finite-dimensional feature map for both kernels, which is exactly what is provided in this paper.

# E  Methods to correct the bias of Tanimoto dot product random features

Here, we describe the bias correction strategies mentioned in Section 4.2 and tested experimentally in section 6. We begin by noting that when $x, x' \in \mathbb{R}^d_{\geq 0}$, $x \cdot x' \geq 0$, and therefore, the power series

$$T_{DP}(x, x') = \sum_{r=1}^{\infty} (x \cdot x')^r (\|x\|^2 + \|x'\|^2)^{-r}$$

is monotone. Therefore, if we use an unbiased sketch for the truncated series

$$\sum_{r=1}^{R} (x \cdot x')^r (\|x\|^2 + \|x'\|^2)^{-r}$$

as the one constructed in Section 4.2, the final sketch will be biased *downward*:

$$\mathbb{E}(\Phi(x) \cdot \Phi(x')) < T_{DP}(x, x')$$

for all $x, x' \in \mathbb{R}^d_{\geq 0}$. Below we introduce two strategies to remedy this issue.

## E.1  Bias correction strategy 1: normalize the features

Empirically, we observe that the highest bias occurs in the diagonal elements of the kernel matrix. As the kernel satisfies $T_{DP}(x, x) = 1$ for all $x \in \mathbb{R}^d$, one possible correction is to *normalize* the sketch to obtain

$$\tilde{\Phi}(x) = \frac{\Phi(x)}{\|\Phi(x)\|}.$$

This remains an oblivious sketch, and ensures that the diagonal of the kernel matrix is estimated exactly, at the expense of possibly introducing some bias in off-diagonal elements of $K$.

## E.2  Bias correction strategy 2: sketch the residual

To simplify the algebra, let $t_{x,y} = \frac{x \cdot y}{\|x\|^2 + \|y\|^2}$. The power series for $T_{DP}$ can then be re-written as:

$$
\begin{aligned}
T_{DP}(x, y) &= \sum_{r=1}^{\infty} (t_{x,y})^r \\
&= \sum_{r=1}^{R} (t_{x,y})^r + (t_{x,y})^{R+1} + \sum_{r=R+2}^{\infty} (t_{x,y})^r \\
&= \sum_{r=1}^{R} (t_{x,y})^r + (t_{x,y})^{R+1} + (t_{x,y})^{R+1} \sum_{r=1}^{\infty} (t_{x,y})^r \\
&= \underbrace{\sum_{r=1}^{R} (t_{x,y})^r}_{k^R} + \underbrace{(t_{x,y})^{R+1} (1 + T_{DP}(x, y))}_{\text{truncation error}}
\end{aligned}
\tag{28}
$$

Critically, the truncation error can be written in terms of the kernel value itself, without any infinite sums. This enables a simple procedure to generate random features for *both* the truncated power series and the remainder:

1. Compute and store $\Phi(x)$ as random features for the truncated kernel $k^R$.
2. Concatenate a single 1 onto $\Phi(x)$ to produce features $\Phi_{+1}(x) = (1, \Phi(x))$ which approximate the kernel $1 + k^R$ using only a single extra dimension. Treat this as approximate random features for the kernel $1 + k$.
3. Compute random features $\Phi_{r+1}(x)$ for $t_{x,y}^{R+1}$.
4. Apply SKETCH to the tensor product $\Phi_{r+1}(x) \otimes \Phi_{+1}(x)$ to obtain $\Delta(x)$, which approximates random features of the truncation error $t_{x,y}^{R+1}(1 + T_{DP}(x, y))$.

5. Concatenate the random features $\Phi(x)$ and $\Delta(x)$, to obtain bias corrected random features $\Phi_{\text{bc}}(x) = \Phi \oplus \Delta(x)$.

Overall, these random features are essentially a concatenation of the random features for the truncated power series with an additional random feature estimate of the truncation error, which is formed using both the random features for the first $R$ terms and the random features for the $(R+1)$th term. A nice property of the procedure above is that it *re-uses* the random features $\Phi(x)$ to estimate the error.

# F  Experimental details and Additional Results

## F.1  Datasets and featurization

The 1000 molecules from GuacaMol are included in our code. The molecules from the DOCKSTRING dataset are available online (`https://github.com/dockstring/dataset`).

Note that when using count fingerprints for $T_{DP}$ we used the *square root* of the counts as the fingerprint. This was done for two reasons:

1. To reduce the norms of the vectors (and thereby increase $\zeta$).
2. To roughly make their interpretation consistent with $T_{MM}$: i.e. the "weight" of a fragment in which occurs $n$ times in the numerator/denominator of $T_{DP}$ is $n$ times the weight of a fragment which occurs once.

## F.2  Details of $T_{MM}$ random features

We use the random hash from Ioffe (2010) in our implementation. To hash vectors in $\mathbb{R}^D$ random variables $r_i, c_i \sim \Gamma(2,1)$, $\beta_i \sim U(0,1)$ are drawn i.i.d. for $i = 1, \ldots, D$, and then the following[3] are computed for all $i$:

$$t_i(x) = \lfloor (\ln x_i)/r_i + \beta_i \rfloor, \tag{29}$$

$$y_i(x) = r_i(t_i(x) - \beta_i), \tag{30}$$

$$a_i(x) = \ln c_i - y_i(x) - \ln r_i, \tag{31}$$

$$i^*(x) = \arg \min_{i=1,\ldots,D} a_i(x), \tag{32}$$

$$h_{r,c,\beta}(x) = \left( i^*(x), t_{i^*(x)}(x) \right). \tag{33}$$

Note that equation 29 uses the convention that $\ln 0 = -\infty$. These variables do not have a clear interpretation in isolation so we refer the reader to Ioffe (2010) for an explanation of why this hashing procedure produces a random hash for $T_{MM}$. The hash itself is formed of 2 integers: $i^* \in \{1, \ldots, D\}$ and $t_{i^*} \in \mathbb{Z}$. This unfortunately means that the number of possible outputs is potentially infinite, which would require us to potentially sample an arbitrarily large vector $\Xi$. To avoid this, we first use python's built-in `hash` function to map this pair of integers to a single integer, then take the result module $2^{12} = 4096$. This allows us to sample a small (finite) vector $\Xi$, and although it introduces a small amount of bias this does not appear to be an issue in practice.

Elements of $\Xi$ are always sampled i.i.d. from either a Rademacher or Gaussian distribution (the distribution should always be clear from context).

## F.3  Additional results for $T_{DP}$ random features

Figure F.1 shows that the error for both polynomial random features from TENSORSKETCH and random features for individual terms in $T'_{DP}s$ power series decreases with $O(1/M)$. Figure F.2 shows the overall error for count fingerprints.

## F.4  Gaussian process training details from section 6.2

Our GP models use a constant mean and Gaussian noise. Specifically, the model of the observed labels $y$ is:

$$f(X) \sim \mathcal{GP}\left(\mu, ak(X,X)]\right)$$

$$y(X) \sim \mathcal{N}\left(f(X), \sigma^2 I\right)$$

Therefore, the GP hyperparameters are three scalars:

- The constant mean, $\mu$

---

[3]Note that the presentation of equations 29–33 differs slightly from Ioffe (2010) who defines $y_i$ and $a_i$ to be the exponential of equations 30/31: we wrote it this way because in practice working in log space avoids numerical stability issues. This is explicitly suggested in their paper.

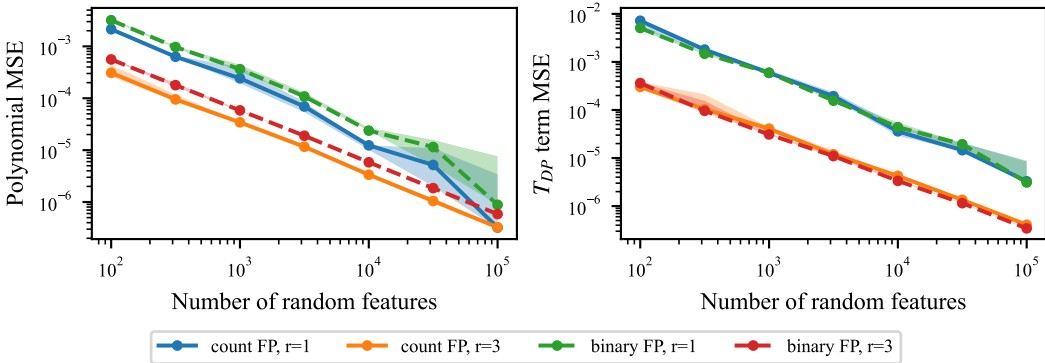

Figure F.1: MSEs for approximating $(x \cdot x')^r$ (**left**) and $((x \cdot x')/(\|x\|^2 + \|x'\|^2))^r$ (**right**) using TENSORSKETCH for various $r$ on count and binary fingerprints as a function of the random feature dimension $M$.

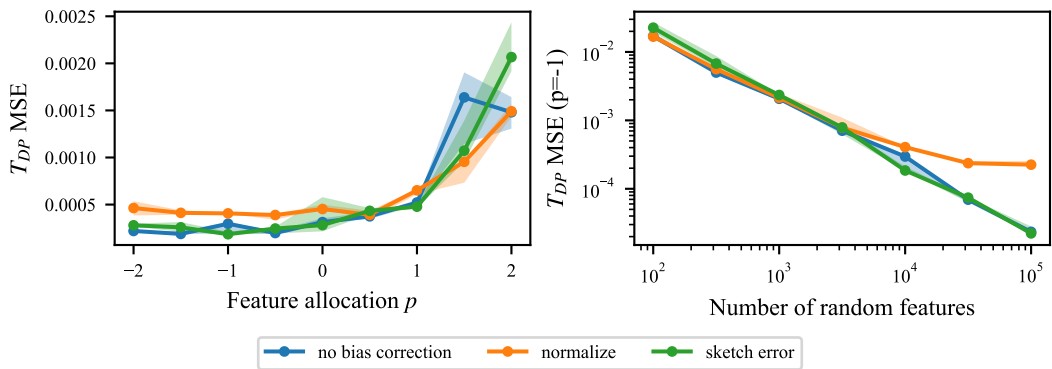

Figure F.2: Same as Figure 3 but with count fingerprints.

- The kernel amplitude/outputscale $a$
- The observation noise $\sigma^2$

GP performance will be greatly affected by the choice of kernel hyperparameters. To ensure that the difference in performance is not due to differences in kernel hyperparameter settings we fit them in a consistent way for all methods. Specifically, for all methods, we start by fitting an exact GP to a random subset of $M = 5000$ data points by maximizing the marginal likelihood with L-BFGS. The different approximations are as follows:

- **Random subset:** `gpytorch`'s exact GP inference is applied on a random subset of $M$ data points (a different subset then was used to fit the hyperparameters).

- **SVGP:** `scikit-learn`'s K-means clustering is run with $M$ clusters to produce an initialization of the inducing points. These inducing points are used to initialize a sparse variational GP (Hensman et al., 2013), implemented in `gpytorch`. The variational parameters are optimized via natural gradient descent with a learning rate of $10^{-1}$ and a batch size of $2M = 10\,000$ for one pass through the dataset. Although the inducing point locations themselves could be further optimized with gradient descent, we chose not to do so for this experiment.

- **RFGP:** First, the training and test sets are converted into $M$ dimensional random features. Then, the posterior equations for inference in Bayesian linear models are used to make predictions on the test set given the training set (Bishop and Nasrabadi, 2006, equations 3.49–3.51). The computation is done in a specific order to avoid forming any $n \times n$ matrices (the Woodbury matrix identity is used extensively for this). This is fairly clearly documented in the code.

Table F.1: Average $R^2$ score for approximate GPs on DOCKSTRING dataset. Attentive FP and MPNN results are taken from García-Ortegón et al. (2022). Other details are the same as Table 1.

| KERNEL | METHOD | ESR2 | F2 | KIT | PARP1 | PGR |
|--------|--------|------|-----|------|-------|-----|
| $T_{MM}$ | RAND SUBSET GP | 0.514±0.002 | 0.810±0.002 | 0.695±0.002 | 0.849±0.001 | 0.426±0.007 |
| | SVGP | 0.578±0.001 | 0.861±0.000 | 0.749±0.000 | 0.889±0.000 | 0.542±0.002 |
| | RFGP (Ξ RAD.) | 0.518±0.002 | 0.838±0.001 | 0.703±0.002 | 0.864±0.001 | 0.465±0.003 |
| | RFGP (Ξ GAUSS.) | 0.517±0.002 | 0.837±0.000 | 0.702±0.001 | 0.864±0.001 | 0.467±0.004 |
| $T_{DP}$ | RAND SUBSET GP | 0.513±0.003 | 0.817±0.001 | 0.696±0.002 | 0.851±0.001 | 0.384±0.011 |
| | SVGP | 0.581±0.001 | 0.865±0.000 | 0.753±0.002 | 0.889±0.000 | 0.543±0.002 |
| | RFGP (PLAIN) | 0.546±0.001 | 0.852±0.001 | 0.716±0.002 | 0.876±0.000 | 0.512±0.002 |
| | RFGP (NORM) | 0.546±0.001 | 0.852±0.001 | 0.715±0.002 | 0.876±0.000 | 0.513±0.002 |
| | RFGP (SKETCH) | 0.545±0.001 | 0.852±0.001 | 0.716±0.002 | 0.876±0.000 | 0.510±0.002 |
| N/A | MPNN | 0.506±0.001 | 0.798±0.005 | 0.755±0.005 | 0.815±0.010 | 0.324±0.096 |
| N/A | ATTENTIVE FP | 0.627±0.010 | 0.880±0.001 | 0.806±0.008 | 0.910±0.002 | 0.678±0.008 |

Note that for $T_{MM}$ it was vital to implement the kernel as

$$T_{MM}(x, x') = \frac{\|x\|_1 + \|x'\|_1 - \|x - x'\|_1}{\|x\|_1 + \|x'\|_1 + \|x - x'\|_1} \tag{34}$$

instead of a naive implementation which follows equation 1. This is because such an implementation requires forming a tensor of shape $N \times M \times d$ when calculating the kernel between $N$ and $M$ points in $\mathbb{R}^d$ (for example $T_{ijk} = \min(x_{ik}, y_{jk})$) which can exceed memory limits for modest $N, M, d$. This identity allows us to use the relatively efficient `torch.cdist` function. Note that we did *not* invent this identity ourselves; we discovered it in Ioffe (2010).

Even with this implementation, $M = 5000$ inducing points did not fit into GPU memory for $T_{MM}$, so all experiments were run on CPU.

### F.5 Metrics and additional results from section 6.2

**Metrics** Table 1 reports *average log probability*, which calculated by first calculating the log probability of each test point individually (i.e. marginally, not jointly), then averaging these values. We also report the coefficient of determination ($R^2$ score), calculated using the function `sklearn.metrics.r2_score`. This measures only the error of the GP mean. A value of 1 indicates perfect prediction, while a value of 0 can be achieved by predicting the sample mean for every data point.

**Additional results** Table F.1 reports the average $R^2$. Trends are similar to Table 1. We also include baselines for two types of graph neural network: Attentive FP (Xiong et al., 2019) and MPNN (Gilmer et al., 2017). Although the performance of GP methods does not match that of Attentive FP, it is often close, suggesting there is potential for approximate GPs to be competitive with graph neural networks for molecular property prediction.

