# OpenReview forum: "Tanimoto Random Features for Scalable Molecular Machine Learning"
_NeurIPS.cc/2023/Conference — NeurIPS 2023 poster_

### Official Review · Reviewer_pMsa · 2023-06-26

**Soundness:** 3 good
**Presentation:** 3 good
**Contribution:** 3 good
**Rating:** 7
**Confidence:** 3

**Summary:**

Random feature map can overcome the limitation of kernel methods, and reduces its computational cost to linear in the number of training data points. Recent advances in random feature map broaden its application to not only to stationary kernels, but also to non-stationary kernels such as product kernels. The authors propose two kinds of random features for Tanimoto coefficient such that kernel methods based on Tanimoto coefficient can be applied to large datasets.

**Strengths:**

1. Proposal of two random features
2. Theoretical analysis on the variance and approximation error for the proposed random features.
3. Demonstrated efficiency through computational experiments.
4. Proposal of positive semidefinite Tanimoto product kernel for real-valued vectors.

**Weaknesses:**

As mentioned by the authors in Discussion,
1. Unexpectedly slow computation of T_MM
2. Uncertainty on the correctness of the error bound

**Questions:**

“2. Uncertainty on the correctness of the error bound”
This reviewer finds this a serious point. What is the reason behind this ? Bound not tight enough, or bug in experiments ?
Please describe what the authors can do to resolve it.

**Limitations:**

The authors clearly mentioned the limitations of their work.

---

> ### Author Rebuttal · Authors · 2023-08-09
>
> Thank you for reviewing our paper. We will try to respond to your questions / concerns below.
>
> > Uncertainty on the correctness of the error bound” This reviewer finds this a serious point. What is the reason behind this ? Bound not tight enough, or bug in experiments ? Please describe what the authors can do to resolve it.
>
> The reviewer may have misinterpreted our comment: we are not uncertain about the correctness of the bound. What we intended to say is:
>
> 1. In practice, the error could be much lower than the bound suggests (the bound might not be tight).
> 2. The bound used a specific set of hyperparameters. However, it is possible that tuning the hyperparameters differently could result in a lower error in practice.
>
> To be clear: we are not uncertain about the correctness of the error bound and we do not believe there is a bug in any of our experiments.
>
> > Unexpectedly slow computation of T_MM
>
> This is not fundamental, it was just because we could not find an easy way to vectorize its computation in python. A good C++ implementation could be fast; we just didn’t do this. The time complexity still scales linearly with N and D.
>
> Please let us know whether this addresses your concerns and whether you have any outstanding questions or concerns.

---

> > ### Comment · Reviewer_pMsa · 2023-08-17
> >
> > Thank you for the response.
> > The rebuttal by the authors clearly resolved my concerns.
> > I would like to raise my score taking into account the authors' response and the discussions between the authors and the reviewers.

---

> > > ### Author Response · Authors · 2023-08-18
> > > **Thank you**
> > >
> > > Thank you for reading our rebuttal and raising your score! We appreciate your contribution to the review process.
> > >
> > > Sincerely,
> > > Authors

---

### Official Review · Reviewer_jyA1 · 2023-06-27

**Soundness:** 4 excellent
**Presentation:** 4 excellent
**Contribution:** 3 good
**Rating:** 7
**Confidence:** 2

**Summary:**

The paper investigates two low-rank approximations for the tanimoto similarity in the context of molecules. The two approximations are both random features newly proposed in the paper using random hashes and sketches. As theory, the paper contributes a proof of optimality for a certain way of constructing one of the approximations and an error bound for the operator norm of the other. They perform four experiments answering the immediate questions about the random features and investigating the claimed advantages of the methods.

Thank you for any time you take to answer my questions!

**Strengths:**

1. They propose a scheme to use random hashes to index an alphabet of random vectors to construct random features as a low rank approximation of the tanimoto similarity. It has several desirable properties in terms of their variance and they prove what the optimal choice is for the alphabet of random vectors.
2. They prove that a previously known kernel is for binary inputs is also a kernel for real vectors. For real vectors their first method of constructing random features does not work and they propose another novel method for constructing random features thus arriving at random features with desirable properties like differentiability which allows for using them in e.g., bayesian optimization. For this they hold significant promise since they are fast to compute as well.
3. The paper provides an error bound on the operator norm of the constructed T_DP random features which apparently is important for sketches to work well according to a citation that I did not check.
4. All claimed advantages are investigated in 3 experiments and the authors do not shy away from showing better performing methods for some tasks and the weaknesses of the random features/kernel methods in general. The experiments are insightful and well chosen.

**Weaknesses:**

1. Main concern: How relevant are hand-crafted kernels, and what are the advantages compared to deep learning embeddings? It seems that also in the DOCKSTRING experiments, deep-learned features fare better. Are there any efficiency and scalability advantages? If the answer is that there are no real advantages, I would see it as a large drawback even if we can say that, in practice hand crafted kernels are still often commonly used. If the answer is otherwise, I think it would be important to discuss this in detail.
2. A more extensive investigation of deep learned fingerprints would be valuable (maybe I am wrong, and you can explain why not): There already is an investigation of deep learned embeddings in terms of Attentive FP in Table 1, but how do deep learning FPs compare in the other experiments, could the f(lambda) computation be extended to deep learned fingerprints?
3. A more extensive investigation of deep learned fingerprints would be valuable (maybe I am wrong, and you can explain why not): If other embeddings from SSL methods, such as those from “Self-supervised Graph-level Representation Learning with Local and Global Structure” would be tried as well that would be interesting or the pretrained features from “Uni-Mol: A Universal 3D Molecular Representation Learning Framework”.
4. Are the baselines in the PMO benchmark actually strong, or are they just easily implemented baselines, and that is why they already were in the benchmark, and one should instead, e.g., compare against other kernels?
5. How do we know that the DOCKSTRING scores reflect/have the same ranking as actual binding affinities, and is the DockString task actually relevant as a real-world benchmark? I think you are making a mistake in saying they are binding affinity tasks - am I wrong in that?
6. Why can we assume that T_MM without random features is the best and should be a baseline to compare to in, e.g., Figure 1? Should a comparison with other kernels not also be considered?
7. Is it fair to say that we now have justification for using T_DP for nonbinary vectors through your Theorem 4.1, but no new capability is unlocked in the sense of the kernel being known and used previously?

**Questions:**

1. Has the trick with random hashes for constructing random features based on an alphabet of random vectors and indexing that with the hashes really never been done before?
2. (Only if you can find the time to answer) Regarding data oblivious sketches: What does it mean for a sketch to be data dependent? Just that it for instance makes assumptions on the sparsity of the kernel matrix? What would be an example of a data-dependent sketch?
3. What is Attentive FP? A half-sentence explanation in the paper might be useful as well.
4. Corollary 4.2 and Theorem 4.1 imply that TDP is an extension of set-valued Tanimoto similarity to real vectors. Is there any inherent value to that or why is the Tanimoto similarity put on such a high pedestal?

**Limitations:**

The paper seems very transparent with weaknesses and does not shy away from e.g. showing results of better-performing methods on certain tasks. They point out limitations in a dedicated section. These are not just conjured up little limitations or remarks about possible small extensions as we see them all too often in ML papers. Instead, they are significant such as computation difficulties that result in speed impairments. The authors could have easily hidden this to make their method seem better but they do not. Thank you!

Maybe the paper should discuss the limitations compared to deep learned fingerprints more and e.g. the weakness compared to attentive FP instead of just showing the results.

---

> ### Author Rebuttal · Authors · 2023-08-09
>
> Thank you for your kind review and very positive rating. We will try to respond to your questions / concerns below.
>
> > How relevant are hand-crafted kernels?
>
> Although hand-crafted kernels are less relevant than a decade ago, they are still very relevant in applications with small datasets (where neural networks overfit) or where interpretability/robustness are important (because their behavior can be theoretically characterized). An example where both of these considerations apply is Bayesian optimization of molecules, wherein predictions and uncertainty estimates of a model are used to choose promising molecules for experiments. In contrast, the conditions of the DOCKSTRING experiment in section 5.3 are very favorable to neural networks (large dataset, only assessing mean prediction accuracy). It is not surprising that the deep learning approach did better here.
>
> We think that “hand-crafted kernels” vs “deep learning” is perhaps the wrong dichotomy, because most kernels (including the ones described in our paper) could be applied to learned fingerprints from neural networks instead of hand-engineered features. We did not explore this angle in our paper since our focus was scalable estimation of these kernels rather than choosing the best features for these kernels. We used fingerprints because this is the most common molecular feature used with Tanimoto similarity.
>
> > A more extensive investigation of deep learned fingerprints would be valuable (maybe I am wrong, and you can explain why not)
>
> We aimed to compare with deep learning methods in situations where it was appropriate. In particular:
> - The experiment in section 5.1 aims to reconstruct matrices of Tanimoto coefficients using random features. The key claim of our work is that the random features we propose can do this, so the goal in this section was to demonstrate this experimentally. Deep learning fingerprints are not trained to do this, and there is no reason to expect that the inner product of deep learning fingerprints would approximate a Tanimoto coefficient (in fact the inner product of deep learning features is usually unbounded). Therefore we did not think this was a reasonable comparison. We think adding it would distract from the main point of the section, which is experimentally verifying the claims of sections 3-4.
> - Deep learning methods were used by Gao et al (2022) as part of the PMO benchmark, and therefore _we did (implicitly) compare against them_ (and Tanimoto GP performed better). We can change the text to emphasize that this comparison was made. The Thompson sampling experiment is a unique capability of methods which make correlated predictions (like GPs), which deep learning methods generally do not. Therefore we could not perform a comparison here.
> - In section 5.3 we compared a scalable kernel method operating on hand-engineered features with an end-to-end deep learning system. We thought this was a sensible comparison because both of these approaches might be used by practitioners. We chose not to include more deep learning methods because it was already clear that the kernel method performs worse (which was the expected outcome).
>
> Therefore we think that it would be difficult to include a thorough investigation of deep learning fingerprints without diluting the overall narrative or significantly expanding the scope of the work. We are however excited to explore applications of scalable Tanimoto kernels in future work, which will involve comparing to deep learning methods on other tasks.
>
> > Are the baselines in the PMO benchmark actually strong?
>
> We think that they are reasonably strong: they included a wide variety of state-of-the-art methods and performed hyperparameter tuning.
>
> > How do we know that the DOCKSTRING scores reflect actual binding affinities, and is the DockString task actually relevant as a real-world benchmark?
>
> We don’t know exactly, but DOCKSTRING targets were chosen to have a reasonably strong correlation between docking score and empirically measured binding affinity. However, we see this package as simply a tool to benchmark our methods against moderately complex molecular properties, and it is one of the few large and publicly-available labeled datasets of small molecules.
>
> > Should a comparison with other kernels not also be considered [in Figure 1]?
>
> The aim of Figure 1 is to evaluate the reconstruction error of various random features approximations of the Tanimoto kernel matrix. This figure does not evaluate the statistical performance of kernel methods for a given molecular prediction task. We would not argue that T_MM should be the definitive baseline in such a task, and we include a limited experimental evaluation of this sort in Sections 5.3.
>
> > Is it fair to say that we now have justification for using T_DP for nonbinary vectors through your Theorem 4.1, but no new capability is unlocked in the sense of the kernel being known and used previously?
>
> I think a new capability is unlocked because even though it was used, it doesn’t look like it was used on non-binary vectors.
>
> > Has the trick with random hashes for constructing random features based on an alphabet of random vectors and indexing that with the hashes really never been done before?
>
> As far as we are aware, no. This is indeed a very natural idea, but we were unable to find a reference to it in the literature.
>
> > What does it mean for a sketch to be data dependent?
>
> A data dependent sketch is one which applies to a specific dataset only. Two examples are Nystrom approximations or inducing point methods.
>
> > What is Attentive FP?
>
> It is a type of graph neural network.
>
> > [...] why is the Tanimoto similarity put on such a high pedestal?
>
> Even though the Tanimoto kernel is a predominant metric in cheminformatics, we see no reason why it should be put on a pedestal. It is simply the kernel which our work focuses on approximating.
>
> Please let us know if this answers your questions and addresses your concerns.

---

> > ### Comment · Reviewer_jyA1 · 2023-08-13
> > **Multiple weaknesses refuted/explained. Raising my score. My confidence remains low.**
> >
> > Thank you for taking the time for the concise and helpful responses.
> >
> > ___
> >
> > Response to weakness 1:
> >
> > The weakness was based on a misunderstanding of mine and no longer has to be considered.
> >
> > ___
> >
> > Response to weakness 2 and 3:
> >
> > With my misunderstanding resolved, I agree with the assessment that the comparison to deep learning methods addresses the relevant questions, and the suggested directions are not the most relevant here.
> >
> > ---
> >
> > > Even though the Tanimoto kernel is a predominant metric in cheminformatics, we see no reason why it should be put on a pedestal. It is simply the kernel which our work focuses on approximating.
> >
> > My comment was regarding *why* you choose Tanimoto similarity. To me it seemed that the paper regards it as the go-to best option and that is, therefore, what we want to approximate. Does your approximation also work for other kernels?
> >
> >
> > ---
> >
> > Multiple of my weaknesses are invalid with the explanations of the authors. I think the paper should be accepted and raise my score. I do not raise it further since my credence still is that it is a solid good paper but not of outstanding impact.

---

> > > ### Author Response · Authors · 2023-08-14
> > > **Thank you for reading our rebuttal & raising your score**
> > >
> > > Thank you very much for reading our rebuttal and raising your score. We want to respond to the points in your most recent comment.
> > >
> > > > My comment was regarding why you choose Tanimoto similarity. To me it seemed that the paper regards it as the go-to best option and that is, therefore, what we want to approximate. Does your approximation also work for other kernels?
> > >
> > > We do not claim that it is the go-to _best_ option: our observation is simply that it is a very popular kernel / distance metric and is often the first thing that chemistry researchers will try. We take its widespread use as evidence that it works well in practice (or at least well enough to be a satisfactory go-to kernel/distance). It is not the only go-to kernel / distance metric though: Euclidean distance and kernels based on Euclidean distance like the RBF kernel are also popular, just slightly less so in the sub-field of chemistry.
> > >
> > > Why then do we focus on Tanimoto? Because, unlike other popular kernels, there were no random feature approximations for scalable kernel estimates. This is the contribution of our work. Essentially it puts Tanimoto on "equal footing" with other kernels (in terms of the features available for these kernels). However, our random features do not work for other kernels: they are just for $T_{MM}$ and $T_{DP}$.
> > >
> > > > my credence still is that it is a solid good paper but not of outstanding impact.
> > >
> > > As the authors we hope that our work is widely used and feel confident that it will at least be used in some chemistry-related problems, but acknowledge it may not be used more broadly, especially in fields which do not already use Tanimoto similarities. We understand and respect your decision to not raise your score further, especially since you say you are not very confident in your judgment.
> > >
> > > Thank you again very much for engaging with our paper!

---

> > > > ### Comment · Reviewer_jyA1 · 2023-08-15
> > > >
> > > > I appreciate the additional time taken to further answer my questions.
> > > > I am glad you understand my decision and also hope that it will be used in chemistry-related tasks and have a real-world impact.
> > > >
> > > > Thank you for the educational paper and informative discussion.

---

### Official Review · Reviewer_A9mq · 2023-07-05

**Soundness:** 4 excellent
**Presentation:** 4 excellent
**Contribution:** 3 good
**Rating:** 7
**Confidence:** 3

**Summary:**

This work develops random feature maps to approximate the Tanimoto coefficient among molecular datapoints. Approximating the Tanimoto coefficients allows this measure to be used as a kernel in classical machine learning approaches on larger datasets such as clustering and Bayesian optimization.

**Strengths:**

The presentation and technical precision of this work is excellent. The authors provide sufficient background to their approach and logic such that even a reader with a rudimentary understanding of analysis and algebra can comprehend their work as well as understand the chain of logical events constituting the work's contribution to research. While much of my research focuses on deep learning, I am deeply appreciative of the authors' work.

It is my opinion that the primary contribution of this work is to develop approximations to the Tanimoto kernel. This enables its usage in classical non-parameteric machine learning algorithms such as bayesian optimization and clustering. The authors present two different approaches---one based on hashing and another on power-series expansion---that each convey disparate benefits and drawbacks. While outside my primary research focus, the solution is quite technically interesting, and I believe it would be relevant for non-deep learning application of molecular science.

The empirical evaluation on binding-affinity prediction is also well-received as it illusatrates how approximating the Tanimoto kernel can be translated into a real-world application.

A final point of appreciation relates to the clarity and honesty of the writing style and evaluation --- the authors do not seek to claim sota or surpass the performance of the most recent deep learning methods, but rather make a fundamental contribution that would be relevant to scaling non-parametric algorithms operating on molecular data to larger datasets. It is also refreshing to read a paper that is very precise with mathematical jargon, using accurate terminology to leverage existing structure in analysis and not proving theorems that are only tangentially related to the thrust of the work.

**Weaknesses:**

With the continued march of progress in deep learning algorithms---and their application to domains of molecular science---it is likely this approach will not see significant stand-alone use in applications. The counterpoint to this estimation is that there's exciting work integrating deep learning with classical non-parametric algorithms, and such approaches may leverage non-parametric approaches as inductive biases to assist representational learning.

**Questions:**

Please explain L156 more: $x,x' \in \mathbb{R}^d = \vec{0} \Rightarrow T_{DP}(\vec{0}, \vec{0}) = 1$? This does not appear to be a continuous function and therefore not differentiable at 0 (which is at odds with the prior paragraph L149 motivating the need for differentiable random feature maps for Bayesian optimization). Perhaps reformulate (6) as a sequence of functions (i.e. \begin{cases}..)  to include 0 and then argue for continuity at 0 ($T_{DP} \rightarrow 1$ as $x,x' \rightarrow 0$).

**Limitations:**

The authors have adequately described the limitations of their approach.

---

> ### Author Rebuttal · Authors · 2023-08-09
>
> Thank you for your kind review and very positive rating. We will try to respond to your questions / concerns below.
>
> > With the continued march of progress in deep learning algorithms---and their application to domains of molecular science---it is likely this approach will not see significant stand-alone use in applications. The counterpoint to this estimation is that there's exciting work integrating deep learning with classical non-parametric algorithms, and such approaches may leverage non-parametric approaches as inductive biases to assist representational learning.
>
> While deep learning has many strengths, particularly on large datasets, we think the approaches in this paper do have their niche:
>
> 1. Kernel methods are generally more useful on small datasets (where neural networks typically overfit). Small datasets of size <1000 are very common in chemistry. Although small datasets generally don’t require approximate kernel methods like the one we present in our paper, one exception to this is approximate Thompson sampling from Gaussian processes which scales cubically in the number of _test_ points (this is desirable for Bayesian optimization). Even with a small training dataset, making predictions on a large test set is expensive. In section 5.2 we show that the random features in our paper are useful for this.
> 2. In instances where reliability or interpretability are important, kernel methods have a clear advantage over neural networks, which often make spurious predictions (e.g. adversarial example)
>
> > Please explain L156 more: ? This does not appear to be a continuous function and therefore not differentiable at 0
>
> $T_{DP}$ is defined in equation 6, but this equation is ill-defined at (0,0). Line 156 gives  the definition of $T_{DP}$ at this point. The choice $T_{DP}(0,0)=1$ ensures that the kernel is positive definite over all of R^n.The kernel is continuous and differentiable everywhere except at (0,0).

---

> > ### Comment · Reviewer_A9mq · 2023-08-14
> >
> > I have read the authors' rebuttal and the other reviews for this work. I maintain my score.

---

### Official Review · Reviewer_b6mP · 2023-07-14

**Soundness:** 4 excellent
**Presentation:** 4 excellent
**Contribution:** 3 good
**Rating:** 8
**Confidence:** 2

**Summary:**

This paper presents two methods for approximating a Tanimoto kernel using random features, shows that the approximations are good, and validates the utility of the proposed approach by applying it to three different settings using molecular fingerprint data.  The writing is clear, and the contributions are clearly stated.  Scaling the Tanimoto kernel to larger datasets is of practical importance.


**Strengths:**

Clarity of exposition.

A nice combination of theoretical and empirical results.

The experiments are extensive and show convincingly that the method works well on a variety of real datasets.

I particularly liked the background in Sections 2 and 3, which set up the contribution of this work very nicely.

The construction of the data-oblivious sketch (Section 4.1) is elegant.


**Weaknesses:**

One of the main theoretical result is really just to show that Equation (6) is positive definite not just on binary inputs (which was shown in 2005) but for real vectors.  This seems like a relatively straightforward result.


**Questions:**

I was confused by the status of Theorem 4.7.  Is this a novel contribution, or just a restatement of something previously shown by (Cohen et al. 2015)?

What is the intuition for why T_{DP} achieves better  accuracy than T_{MM} in Section 5.3?

---

> ### Author Rebuttal · Authors · 2023-08-09
>
> Thank you for your kind review and very positive rating. We will respond to the negative points and questions from your review below.
>
> > One of the main theoretical result is really just to show that Equation (6) is positive definite not just on binary inputs (which was shown in 2005) but for real vectors. This seems like a relatively straightforward result.
>
> The proof that $T_{DP}$ is positive-definite is indeed straightforward, although we would highlight that our proof technique is novel, and in fact much more concise than previous proofs of positive definiteness for binary inputs. We do however agree that it is not the most technically involved result of our paper (that would be theorem 4.7).
>
> > I was confused by the status of Theorem 4.7. Is this a novel contribution, or just a restatement of something previously shown by (Cohen et al. 2015)?
>
> This was a novel theorem. We mention Cohen et al (2015) because that paper pioneered the study of approximate matrix multiplication in terms of stable rank, and because it gives a general bound for the error of approximate matrix multiplication, which sets the standard of what kind of error bound is considered “good” for this type of problem. However, the specific problem studied in Cohen et al is matrix multiplication, whereas our paper focuses on _creating_ random feature matrices for the Tanimoto kernel.
>
> > What is the intuition for why $T_{DP}$ achieves better accuracy than $T_{MM}$ in Section 5.3?
>
> We don’t really know. It is hard to say why one kernel performs better than another kernel on any specific dataset. It could also be related to the hyperparameter tuning for the inducing point optimization.

---

> > ### Comment · Reviewer_b6mP · 2023-08-10
> >
> > I have read the authors' rebuttal as well as the other reviews. I am satisfied with the responses, and I generally think the response to the other authors are clear and to-the-point.

---

### Official Review · Reviewer_stq5 · 2023-07-20

**Soundness:** 3 good
**Presentation:** 2 fair
**Contribution:** 2 fair
**Rating:** 5
**Confidence:** 3

**Summary:**

The paper derives two random feature maps for Tanimoto kernels of fingerprint data. The feature maps are applied in GP and BO settings to allow accelerated primal learning.

**Strengths:**

The idea of featurising Tanimoto is clever and novel. The two feature maps are seemingly well derived, and theoretical analysis is provided. The paper shows how these ideas open the door for feature-based learning in GP setting, which is potentially significant.

The results demonstrate the efficiency of the feature maps, and show good results when applied to BO/GP problems.

**Weaknesses:**

The paper clarity is frustratingly low, and it lacks a lot of rigor and its claims or contributions are not very clear.

The paper is vague about the cost or running times of the new feature maps, and how they relate to exact kernel computation.

Computation of Tanimoto is just few matrix products, and should be fast. I don’t think the paper has demonstrated sufficiently what was the bottleneck earlier in using Tanimoto, or why do we need these new feature maps in the first place. Furthermore, if we have relatively small fingerprints and linear kernels, why do we even need GPs at all: wouldn’t a Bayesian linear regression do the same thing and avoid any need to construct feature maps? Why don’t we just use the datapoints as feature vectors directly? It seems that the paper is motivated by plugging Tanimoto’s into existing (and complex) GP frameworks, without first dismissing simpler direct approaches.

**Questions:**

- “In the early stage..”. Not sure I agree with this statement. Big datasets are used in all stages of drug development. It’s also not clear why deep learning methods struggle, and there is no citation. Can you explain?
- \cdot is undefined
- What does random function mean? Does it change every time? Is it stochastic? Is it random? Random in what sense? What is a random feature map? The presentation lacks rigor.
- What does eq 2 mean? What does E_f mean? What is the measure or probability of f? It seems that we take infinitely many feature maps, and require that they together average to a kernel’s feature map. This feels strange, where is this coming from? It does not seem to be RFFs, since in RFFs the expectation is over frequencies (ie taking m to infinity), not functions. Eq 2 doesn’t require that any particular f would approximate the correct kernel value, and thus the statement hatK = f.f is not true based on eq 2.
- “There is no general formula to define random features for non-stationary kernels”. Not sure this is correct. Please check seminal papers Kom-Samo et al “Generalised spectral kernel”, and Shen et al “Harmonizable mixture kernels with variational Fourier features”; and followup works.
- P_h is undefined
- What is “discrete range”?
- What is an “independent copy”? So we copy the number independently, resulting in just identical copies (surely all copies are independent)? Or is the idea that each is an iid sample of the random variable? Is \xi scalar?
- What does \Xi_h(x) mean? I don’t understand where the h(x) goes. I think h(x) is a selector function that gives one choice out of K choices, and then we pick that \xi from the list of \xi’s? But that would be just \xi_{h(x)}, so \Xi_h(x) should then be something else. Maybe it’s a set of K \xi’s that we somehow index? It would help a lot if you would describe the domain and size of all symbols and variables in the paper (eg. is \phi(x) a scalar or vector?). It’s also annoying that \cdot’s are used to what are seemingly scalar products (line 136, or maybe its not scalar? Ok, line 144 declares them as scalars.)
- What is the E_\Xi,h expectation over? Are there measures over \Xi and h? What are they?
- What is a “tight” lower bound?
- Theorem 3.1. states that \xi needs to have variance 1. Yet, the optimal \xi is declared to be Unif(-1,1), which has variance of 1/3. This seems to nullify the entire Rademacher result. Can you explain?
- Why is the variance at most 1/m?
- I’m confused what the 1/m analysis actually says. Let’s assume we have million-dimensional input vectors. Now we seemingly only need 1000 features to accurately represent them. Surely this can’t be true. What if we have trillion input dimensions? I think this analysis has some caveats that are not made transparent. Can you explain? Furthermore, the eq 4 requires us to average both \Xi and h’s. It seems that the text is forgetting the \Xi part. I’m also now realising that I don’t understand what the “m” random features refer to. Is this m hashes, or m \xi’s, or m \Xis, or something else?
- I don’t understand what the 1-bit remark means. What is an “entry” and how come it needs only one bit? Why do we get 100kB?
- I don’t understand what is the contribution in sec 3. It seems that a new feature map is introduced. Ok, but how is it novel? What problem does it solve? What is its significance? How does it improve over earlier stuff? The text mentions low-variance, but there has not been any discussion of variance of earlier methods. How do we know the variance is then lower?
- Sec 3 also feels vague: the hash has not been described. Surely the hash affects things and can’t be ignored? What if we have a really crappy hash that outputs constant value: surely then the line 136 would not be true. Please make all assumptions explicit.
- Why eq 7 doesn’t support x=x’=0? The equation seems to work for it as well.
- Are m_r related to each other?
- I don’t understand line 194. So phi is the feature map of kernel (x+x’)^-r, but why does the feature inner product add squares into the kernel? This feels surprising. Why do we even care about the non-square kernel, line 190 doesn’t have that either.
- It’s not clear what u or i mean conceptually. What is their role?
- The story seems to restart at 4.2. Suddenly we are talking about data matrix, it’s linear kernel and its rank. I can’t follow what’s happening. Is a paragraph missing between 4.2. and end of 4.1.? Can you explain? It’s also strange that we redefine data as A, but later keep using datapoints as x, and also X appears (whatever it is, maybe data again?).
- What does min_i |x_i|^2 mean? I can’t follow the notation. So it seems that we take the norm of the data vector, which gives a scalar, and then its minimum is just the scalar itself. Or is this the smallest element in x_i? Is x_i a vector or an element of a vector? It would be helpful to use boldfacing on vectors.
- What does zeta mean? In usual fingerprints we have always some zero values, which would indicate that zeta is zero. But then the eq 204 seems to get screwed up. Does this mean that the theory of sec 4 requires the fingerprints to be strictly non-zero or strictly positive? That would be a significant limitation.
- What is _op?  What is poly(n)? What is tildeO? What is tilde\Omega?
- It seems that the paper hides the actual method description to the appendix, or borrows the TreeSketch/SRHT/OSNAP methods to do the heavy lifting. This makes it a bit difficult to follow what’s going on. Could you summarise the main ideas of these, and how does your method extend them?
- In 5.2. I wonder what is the computational cost between T_DP exact and approx? The exact is just few matrix products, while the approx is something more involved. Why is there such a big running time difference?
- In table 1 are the methods approx or exact?

**Limitations:**

No issues

---

> ### Author Rebuttal · Authors · 2023-08-09
>
> Thank you for your thorough review and thoughtful comments. We appreciated that you found our work to be “clever and novel” and recognized that it “open[s] the door for feature-based learning in [the] GP setting, which is potentially significant.” Due to the 6000 character limit for the rebuttal we will try to answer all the _general_ questions/concerns below, then post specific answers to all questions as official comments.
>
> **Concern 1:** clarity of notation
>
> We agree that we could make some improvements here. Based on the reviewer’s questions, we propose to make the following notation changes:
>
> 1. We will use superscripts to denote vectors in a set and subscripts to denote indices of a vector. Previously there was some ambiguity because they were both denoted with subscripts.
> 2. We will make the definition of random variables and expectations more explicit throughout.
> 3. Keep computational complexity terms like poly(n), but define them in the main text rather than referring the reader to the appendix.
>
> **Concern 2:** necessity of random features
>
> Due to the page limit we gave only a brief introduction to random features in sections 1-2. We realize this may not be enough to help readers unfamiliar with random features to understand their advantages and disadvantages. The key points here are:
>
> 1. It is an approximate method which sacrifices accuracy to improve the time / memory complexity of model training/inference for large datasets.
> 2. Given a dataset of size N, it approximates the kernel matrix $K\approx\Phi \Phi^T$, where $K$ has shape $N\times N$ and $\Phi$ has shape $N\times M$. $M$ controls the quality of the approximation (larger is better, but more expensive). If $M < N$ then it will be faster than an exact kernel method.
> 3. Computing $\Phi$ depends only linearly on $N$ (but could have any dependence on M and the input dimension D).
> 4. This generally reduces the complexity of operations to be linear in N instead of quadratic or cubic. For example, GP inference is reduced from O(N^3) to O(NM^2) (assuming $M < N$).
>
> Therefore, even though the computation of Tanimoto is “just a few matrix products” as the reviewer points out, the bottleneck is high time/memory complexity when N is large.
>
> **Concern 3:** why not use linear models
>
> Although using a linear kernel and doing Bayesian linear regression is another way to avoid high computational complexity, this model is less expressive than Tanimoto. For example, with D-dimensional inputs a GP with a linear kernel can only interpolate through D data points (regardless of their spacing) while a GP with a Tanimoto kernel could interpolate through an arbitrary number of data points provided they are spaced sufficiently far apart.
>
> **Concern 4:** running times of the new features maps
>
> We did not state this because it depends on specific choices of hyperparameters. For $T_{MM}$ the cost is $O(NMD)$ if the hash of Ioffe (2010) is used. For $T_{DP}$, the cost is at most $O(NM^2 D)$. However, please note that:
> - This cost is _additive_, not multiplicative with the inference cost above (so the overall cost is still linear in N)
> - Unlike kernel inference, computing $\Phi$ can be done in parallel for different data points, so a high computational cost does not necessarily imply a long runtime.
> We will add this information to the main text.
>
> **Concern 5:** novelty / improvement of random features over previous work
>
> To clarify, we are not aware of any prior works proposing random features for the Tanimoto kernel. We view them as the “first of their kind” rather than an improvement of prior work. Of course, to develop these features we leveraged innovations and developments of prior papers, notably random hashes and TreeSketch.
>
> **Concern 6:** general spectral random features for non-stationary kernels
>
> We were not aware of the works by Kom-Samo et al and Shen et al: thank you for pointing them out. These works are based on a generalization of Bochner’s theorem for harmonizable, non-stationary kernels. Although the approach does seem broadly applicable, we noticed the following limitations:
>
> 1. Although the theorems in these papers  guarantee the existence of a spectral representation  for harmonizable kernels, this may not be easy to derive or indeed take a simple form for a given kernel.
> 2. The representation does not lead in general to a random features approximation (See the discussion in Section 2.2.1 of Shen’s dissertation: Spectral Kernels for Gaussian Processes).
> 3. One could attempt to approximate the Tanimoto kernel with the sparse mixture kernel by Kom-Samo and Roberts or the harmonizable mixture kernel of Shen et al. These approximations sometimes lead to random features. However, even though these mixture kernels are dense in the space of non-stationary kernels, there are no results quantifying how many mixture components would be necessary to get a reasonable approximation.
>
> Despite these limitations, we think this is a very relevant suggestion which is worth investigating in the future. We will add citations to this literature in the final version of the paper.
>
> **Summary:** We hope that this response and the associated comments answering your questions have adequately addressed all of the reviewer’s concerns. Please let us know if you have any other questions or concerns.

---

> > ### Author Response · Authors · 2023-08-10
> > **Answers to reviewer questions [1/N]**
> >
> > As promised, here are detailed answers to the specific questions raised in your review. Kindly refer to our rebuttal for our more general response.
> >
> > > “In the early stage..”. Not sure I agree with this statement. Big datasets are used in all stages of drug development. It’s also not clear why deep learning methods struggle, and there is no citation. Can you explain?
> >
> > We agree that it is an overgeneralization to say that “only small datasets are available” in early-stage drug discovery, but in general the amount of data decreases as the label relevance increases [1]. For a given drug discovery project there may be billions of unlabelled molecules, millions of labels from virtual screening, tens of thousands of labels from high-throughput experiments, but only a handful of in-vivo experiments. This makes small data problems quite common, especially if a scientist judges labels from larger datasets to be unreliable or irrelevant. Furthermore, deep learning methods are widely-known to overfit on very small datasets [2].
> >
> > We are happy to add these citations.
> >
> > [1] Bender, Andreas, and Isidro Cortés-Ciriano. "Artificial intelligence in drug discovery: what is realistic, what are illusions? Part 1: Ways to make an impact, and why we are not there yet." Drug discovery today 26.2 (2021): 511-524.
> >
> > [2] Brownlee, Jason. Better deep learning: train faster, reduce overfitting, and make better predictions. Machine Learning Mastery, 2018.
> >
> > > \cdot is undefined
> >
> > It means inner product (Euclidean scalar product). We believe this is standard notation in the field.
> >
> > > What does random function mean? Does it change every time? Is it stochastic? Is it random? Random in what sense? What is a random feature map? The presentation lacks rigor.
> >
> > Here a _random function_ is a random variable which is function-valued. A _random feature map_ is a random function which maps from the input space to Euclidean space. Sometimes, random feature maps are parametric functions with parameters which are random variables. We will aim to give more explicit definitions of all probabilistic objects in the final version of the paper.
> >
> > > What does eq 2 mean? What does E_f mean? What is the measure or probability of f? It seems that we take infinitely many feature maps, and require that they together average to a kernel’s feature map. This feels strange, where is this coming from? It does not seem to be RFFs, since in RFFs the expectation is over frequencies (ie taking m to infinity), not functions. Eq 2 doesn’t require that any particular f would approximate the correct kernel value, and thus the statement hatK = f.f is not true based on eq 2.
> >
> > It means expectation with respect to the random function $f$.
> > Although the presentation may appear to be very different from RFFs, it is completely compatible. Consider the class of functions $f(x) = a \sin{\omega x} + b\cos{\omega x}$ for any $a,b,\omega$. Works with RFFs tend to present the random feature map $f$ as a parametric function, with parameters $a,b,\omega$, which are random variables. An expectation with respect to the parameters of this class of functions (mainly the frequency $\omega$) is equivalent to an expectation with respect to the random function $f$.
> >
> > You are right that Eq 2 does not require that any particular f accurately approximates the correct kernel value, we do not claim that $K=\hat{K}$: it is just an approximation. However, it has the nice property of being correct _on average_. This means that averaging many such functions will approach the correct kernel.
> >
> > > “There is no general formula to define random features for non-stationary kernels”. Not sure this is correct. Please check seminal papers Kom-Samo et al “Generalised spectral kernel”, and Shen et al “Harmonizable mixture kernels with variational Fourier features”; and followup works.
> >
> > Thanks again for pointing this out: we addressed this in our main rebuttal. We will change the wording of our statement to acknowledge this.
> >
> > > P_h is undefined
> >
> > P means probability, and h is a hash function sampled randomly from a distribution over hash functions (which we did not define explicitly). As a whole, the left side of equation 3 means “probability that the output of h matches for two inputs x and x’ when h is sampled randomly from p(h)”.
> >
> > > What is “discrete range”?
> >
> > This just means that the set of values the function can output is discrete. Think of it as outputting an integer (although technically it could output a letter or some other discrete object).
> >
> > See additional replies for additional answers.

---

> > ### Author Response · Authors · 2023-08-10
> > **Answers to reviewer questions [2/N]**
> >
> > As promised, here are detailed answers to the specific questions raised in your review. Kindly refer to our rebuttal for our more general response.
> >
> > > What is an “independent copy”? So we copy the number independently, resulting in just identical copies (surely all copies are independent)? Or is the idea that each is an iid sample of the random variable? Is \xi scalar?
> >
> > We say that $X_1,\dots,X_n$ are _independent copies_ of $X$, if they are a sequence of random variables which are independent, each of which is identical to $X$ in distribution. In probability, “independent copy” is a relatively standard term for this. We will happily change this. Also yes, $\xi$ is a scalar.
> >
> > > What does \Xi_h(x) mean? I don’t understand where the h(x) goes. I think h(x) is a selector function that gives one choice out of K choices, and then we pick that \xi from the list of \xi’s? But that would be just \xi_{h(x)}, so \Xi_h(x) should then be something else. Maybe it’s a set of K \xi’s that we somehow index? It would help a lot if you would describe the domain and size of all symbols and variables in the paper (eg. is \phi(x) a scalar or vector?). It’s also annoying that \cdot’s are used to what are seemingly scalar products (line 136, or maybe its not scalar? Ok, line 144 declares them as scalars.)
> >
> > I think this is an instance of our vector vs scalar notation being unclear. As we stated in our general response, we will change this to be clearer. Your interpretation of $h(x)$ as a selector function is correct. $\Xi_h(x)$ is the element of the vector $\Xi$ at index h(x).
> >
> > > What is the E_\Xi,h expectation over? Are there measures over \Xi and h? What are they?
> >
> > It is an expectation with respect to the elements of $\Xi$ and the hash function h. \Xi is a vector of iid samples from p(\xi) and h is sampled from p(h), which was defined implicitly in equation 3 (see our responses above). The expectation written out fully is $E_{h\sim p(h), \xi_1\sim p(\xi),  \xi_2\sim p(\xi), \ldots,  \xi_K\sim p(\xi)}$
> >
> > > Theorem 3.1. states that \xi needs to have variance 1. Yet, the optimal \xi is declared to be Unif(-1,1), which has variance of 1/3. This seems to nullify the entire Rademacher result. Can you explain
> >
> > In theorem 3.1 when we wrote “uniform in $\{-1,+1\}$ we meant it is -1 with 50% probability and +1 with 50% probability, not uniformly distributed in the interval -1 to +1. We were trying to follow the notation of using $[(/)]$ symbols to denote intervals and {} symbols to denote an explicit set, which is why we wrote $\{-1,+1\}$ instead of $[-1,+1]$.
> >
> > > What is a “tight” lower bound?
> >
> > Tight means that the bound is actually achieved, and therefore cannot be any higher: it is the strictest possible bound. By contrast, we could have stated that the variance is $\geq0$, which is a correct bound but not tight because for some inputs the variance is much higher than 0.
> >
> > > Why is the variance at most 1/m?
> >
> > Equation 5 implies that the variance is at most 1, and the variance of the average of $m$ iid random variables is the variance of 1 variable divided by m, leading to a bound of $1/m$.
> >
> > > Sec 3 also feels vague: the hash has not been described. Surely the hash affects things and can’t be ignored? What if we have a really crappy hash that outputs constant value: surely then the line 136 would not be true. Please make all assumptions explicit.
> >
> > We think the confusion here may come from a misunderstanding of equation 3. Hopefully our response above clarifies this. The reviewer’s intuition that the hash can’t be ignored is correct, and indeed, a trivial hash that outputs a constant would not work. Theorem 3.1 requires that the random hash satisfies equation 3, and most hashes, including said trivial hash, will not satisfy this property.
> >
> > See additional replies for additional answers.

---

> > ### Author Response · Authors · 2023-08-10
> > **Answers to reviewer questions [3/N]**
> >
> > As promised, here are detailed answers to the specific questions raised in your review. Kindly refer to our rebuttal for our more general response.
> >
> > > I’m confused what the 1/m analysis actually says. Let’s assume we have million-dimensional input vectors. Now we seemingly only need 1000 features to accurately represent them. Surely this can’t be true. What if we have trillion input dimensions? I think this analysis has some caveats that are not made transparent. Can you explain? Furthermore, the eq 4 requires us to average both \Xi and h’s. It seems that the text is forgetting the \Xi part. I’m also now realising that I don’t understand what the “m” random features refer to. Is this m hashes, or m \xi’s, or m \Xis, or something else?
> >
> > Although it seems surprising and unintuitive, the variance of the scalar random features in equation 4 actually does not depend on the input dimension. This is effectively an inherited property of the random hash function. If it seems “too good to be true”, consider the following points:
> > 1. You are not fully representing high-dimensional vectors, just their Tanimoto coefficient, which is a form of normalized overlap. If the vectors overlap significantly then one only needs to represent this fact, and if they do not overlap then only this needs to be represented. Put another way, you cannot recover much about the underlying vectors from their Tanimoto coefficients.
> > 2. The range of the Tanimoto coefficient is bounded between 0 and 1 (independent of the input dimension)
> > 3. The outcome of equation 3 is binary (either the hash values match or they do not match). Any binary outcome has a Bernoulli distribution with variance at most 0.25, no matter how complex the underlying process which generates the binary outcome.
> >
> > $m$ is the dimension of the vector random features, formed by concatenating m independent scalar random features from equation 4 (and normalizing by sqrt(m)). Since the scalar random features do not depend on the input dimension, neither do these, so their variance is at most 1/m regardless of the input dimension.
> >
> > > I don’t understand what the 1-bit remark means. What is an “entry” and how come it needs only one bit? Why do we get 100kB?
> >
> > I think this should be clarified by our answer above saying that $\xi$ are not uniformly distributed in $[-1,+1]$ but rather -1 with 50% probability and +1 with 50% probability. This means that each element of $\Xi$ can be represented with 1 bit, so if a hash can output 10^6 distinct values then one sample from $p(\xi)$ can be stored for each value using only ~100 kilobytes of memory.
> >
> > > I don’t understand what is the contribution in sec 3. It seems that a new feature map is introduced. Ok, but how is it novel? What problem does it solve? What is its significance? How does it improve over earlier stuff? The text mentions low-variance, but there has not been any discussion of variance of earlier methods. How do we know the variance is then lower?
> >
> >
> > We tried to explain the significance of this in our overall rebuttal. To recap:
> > - It is novel because nobody has introduced a similar feature map in a past paper
> > - It solves the problem of a dataset-independent low-rank approximation to the Tanimoto kernel matrix
> > - Its significance is approximating the elements of this matrix without statistical bias and low variance. Here, “low” refers to low relative to other estimators for similar things (e.g. random Fourier features) rather than previous random features for the Tanimoto kernel, because as far as we are aware there are none.
> > - It does not “improve over earlier stuff” because as far as we are aware there is no earlier stuff. There are analogous methods for other kernels (e.g. RFFs) but these solve a different problem (approximating a different kernel matrix)
> >
> > > Why eq 7 doesn’t support x=x’=0? The equation seems to work for it as well.
> >
> > Substituting these values would lead to terms like (0^2)/(0^2) which are not well-defined.
> >
> > > Are m_r related to each other?
> >
> > There is no fixed relationship. This sketch is essentially a concatenation of sketches for different values of r, each of which has a dimension m_r. The values can be anything, although the choice of m_r will affect the overall accuracy of the approximation.
> >
> > > I don’t understand line 194. So phi is the feature map of kernel (x+x’)^-r, but why does the feature inner product add squares into the kernel? This feels surprising. Why do we even care about the non-square kernel, line 190 doesn’t have that either.
> >
> > The equation on line 190 is incorrect: it should read $(|x|^2+|x'|^2)^{-r}$ (i.e. there should be squares in both). This was just a typo; thank you for bringing it to our attention
> >
> > See additional replies for additional answers.

---

> > ### Author Response · Authors · 2023-08-10
> > **Answers to reviewer questions [4/4]**
> >
> > As promised, here are detailed answers to the specific questions raised in your review. Kindly refer to our rebuttal for our more general response.
> >
> > > It’s not clear what u or i mean conceptually. What is their role?
> >
> > I think this confusion comes from the ambiguity about the notation for elements of a vector. The equation describes an $M$ dimensional sketch for $(|x|^2+|x'|^2)^{-r}$. $u$ is a random variable that specifies the random function (similar to $\Xi,h$ in section 3) and $i$ just refers to the $i$th dimension of the $M$ dimensional output.
> >
> > > The story seems to restart at 4.2. Suddenly we are talking about data matrix, it’s linear kernel and its rank. I can’t follow what’s happening. Is a paragraph missing between 4.2. and end of 4.1.? Can you explain? It’s also strange that we redefine data as A, but later keep using datapoints as x, and also X appears (whatever it is, maybe data again?).
> >
> > In section 4.2 we present an error bound for the random features from section 4.1 which depends on the stable rank of the underlying kernel matrix. We include a definition of stable rank to make this bound understandable. The matrix “A” is just a dummy variable used to define the stable rank. We will eliminate the phrase calling it a “data matrix” on line 216, since this is indeed a bit confusing.
> >
> > > What does min_i |x_i|^2 mean? I can’t follow the notation. So it seems that we take the norm of the data vector, which gives a scalar, and then its minimum is just the scalar itself. Or is this the smallest element in x_i? Is x_i a vector or an element of a vector? It would be helpful to use boldfacing on vectors.
> >
> > It means the smallest square norm of any vector $x_i$ in the dataset. This will be resolved more clearly when we change our notation (see main response). Sorry about this.
> >
> > > What does zeta mean? In usual fingerprints we have always some zero values, which would indicate that zeta is zero. But then the eq 204 seems to get screwed up. Does this mean that the theory of sec 4 requires the fingerprints to be strictly non-zero or strictly positive? That would be a significant limitation.
> >
> > $\zeta$ is the square of the ratio between the fingerprint in the dataset with the largest norm and the fingerprint in the dataset with the smallest norm. There is no requirement that fingerprints be strictly positive.
> >
> > > What is _op? What is poly(n)? What is tildeO? What is tilde\Omega?
> >
> > _op refers to the operator norm (https://en.wikipedia.org/wiki/Operator_norm). $\tilde{O}$ and $\tilde{\Omega}$ are like standard O and $\Omega$ from big-O notation except ignoring poly-logarithmic factors (see https://en.wikipedia.org/wiki/Big_O_notation#Extensions_to_the_Bachmann%E2%80%93Landau_notations). We define these terms in appendix A. We can put a pointer to this at the start of section 4.2.
> >
> > > It seems that the paper hides the actual method description to the appendix, or borrows the TreeSketch/SRHT/OSNAP methods to do the heavy lifting. This makes it a bit difficult to follow what’s going on. Could you summarise the main ideas of these, and how does your method extend them?
> >
> > We tried to summarize these methods in Section 2 and then explain our extension to them in Section 4. A concise definition of TreeSketch is given in Appendix B. We did not intend to hide the actual method but rather present the key idea & contribution in the main text and move the technical details and proofs to the appendix.
> >
> > The main idea is that we create random features for each term in equation 7 separately. We create it by creating random features for $(x\cdot x’)^r$ using TreeSketch with SRHT or OSNAP as the base sketch (essentially an established method) and create features for $(|x|^2+|x'|^2)^{-r}$ with the (novel) estimator in section 4.1. We then combine these random features using an instance of TreeSketch, in a similar way that TreeSketch combines linear kernel sketches to create a sketch of the polynomial kernel (which is a product of linear kernels).
> >
> > > In 5.2. I wonder what is the computational cost between T_DP exact and approx? The exact is just few matrix products, while the approx is something more involved. Why is there such a big running time difference?
> >
> > The difference in running time is because T_DP exact represents exact Thompson sampling on $N$ points (which costs $O(N^3)$) while T_DP approx represents approximate Thompson sampling using the method of Wilson et al which only scales with $O(NM)$. It is not the difference in calculating elements of the kernel matrix itself, for which the exact T_DP is likely to be faster.
> >
> > > In table 1 are the methods approx or exact?
> >
> > They are approximate GPs using the inducing point method. That is what is meant in the Table caption by the word “sparse”.
> >
> > This answers all your questions. Please let us know if anything is unclear or if you have additional questions.

---

> > ### Comment · Reviewer_stq5 · 2023-08-11
> > **resp**
> >
> > Thanks for the lengthy responses. The technical issues were resolved, but I still have some confusion about the motivation of the method.
> >
> > Usually RFFs are applied to linearise non-linear kernels (eg. Gaussian) with complex or non-trivial feature maps. But in Tanimoto the kernel is effectively (the norm scaling does complicate things, but I treat this as a minor technical detail) linear, and thus the feature map should also be just linear wrt the original fingerprint vectors. So why do we need to do RFF at all? Why not just use the original features directly as your (identity, or scaled) feature map?
> >
> > I'm also confused about the remarks about linear regression. GPs are exactly equivalent to Bayesian linear regression in the (kernel) feature space (Rasmussen 2006, chapter 2.1/2.2). I don't see why BLR is "less expressive". There is no issue with "interpolating through only D points". The BLR or LR will learn a function in D-dimensional input space that aggregates the entire data into the posterior.
> >
> > Looking forward to your comments.

---

> > > ### Author Response · Authors · 2023-08-12
> > > **Motivation for our work: non-linear kernels are more expressive than linear kernels and Tanimoto kernel cannot be easily linearized**
> > >
> > > Thanks for your prompt response! We will try to respond to your points about Tanimoto kernel being similar to a linear kernel and the expressiveness of linear regression.
> > >
> > > **First,** we completely agree with the reviewer that "GPs are exactly equivalent to Bayesian linear regression in the (kernel) feature space". However, this means that they _only as expressive as a linear model in that feature space_. Recall that in $\mathbb{R}^d$ one is only able to draw a hyperplane through $d$ points (assuming the hyperplane must pass through the origin), and therefore can only expect a linear model to perfectly interpolate up to $d$ points, regardless of their spacing. This is why most popular kernels, such as the RBF kernel $\exp(-\|x-x'\|^2)$, have an _infinite_ dimensional feature space: this potentially allows methods using these kernels to interpolate through arbitrarily many points if they are sufficiently far apart (even for small $d$). In essence, the "point" of kernel methods is to implicitly use high/infinite dimensional linear models without instantiating their weights. Hopefully this answers your question about GPs vs BLR.
> > >
> > > **Second,** we agree that the Tanimoto kernel is a kind of normalized linear kernel but do not consider this a "minor technical detail": it actually completely changes the structure of the kernel. Although we understand the intuition that the kernel should have a nearly-linear feature map because it is closely related to the linear kernel, we in fact developed a proof that this is not the case. Specifically, we can show that no finite-dimensional feature map exists for $T_{MM}$ across its domain. Although we could not prove that $T_{DP}$ has the same property, we conjecture that it does (and present some intuition for this). To make this response easier to read, we formally state this proposition and present the proof at the end of this response. Note that we just developed this claim/proof yesterday: it was not in our original paper.
> > >
> > > What does this mean? If a finite-dimensional _exact_ feature map does not exist, then the best one can hope for is a finite-dimensional _approximate_ feature map. This is precisely what we present in our paper for $T_{MM}$ and $T_{DP}$.
> > >
> > > **In summary,** our response to the reviewer's questions for our motivation is 1) despite appearing nearly linear, our kernels are actually very non-linear and lack a simple feature map, and 2) we more generally seek to use non-linear kernels to implicitly use a high-capacity, infinite-dimensional linear model instead of a less expressive finite-dimensional linear model. Please let us know if you find this motivation convincing or if you have any additional questions.
> > >
> > > -----
> > >
> > > **Proposition** there does not exist a finite-dimensional feature map for $T_{MM}$
> > >
> > > **Proof**: we will use a proof by contradiction. The setup for the contradiction is as follows: suppose there existed an exact feature map $f:\mathbb{R}^d\mapsto\mathbb{R}^M$ for $T_{MM}$ such that $T_{MM}(x,y)=f(x)\cdot f(y)\quad\forall x,y\in\mathbb{R}^d$. This would imply that any kernel matrix between $N$ points $X$ could be written as an inner product: $T_{MM}(X,X)=f(X)^T f(X)$. Because $f$ outputs $M$ dimensional vectors this would imply the resulting matrix has rank at most $M$. Therefore, under this hypothesis it should not be possible to form a kernel matrix of more than $M$ inputs which is full-rank (invertible). Our contradiction will be to construct such a matrix.
> > >
> > > We now present a way to construct a full-rank $T_{MM}$ kernel matrix with any number of points $N$. Essentially we tried to find a set of points whose kernel matrix resembles the identity matrix, which is full-rank. Pick $a\in(0,1)$, and consider the set of $N$ points in $\mathbb{R}$ $1, a, a^2, \ldots, a^{N-1}$. Denoting the $i$th point in this sequence as $x_i$, it is easy to check that $T_{MM}(x_i, x_j)=a^{-|i-j|}$, and therefore the kernel matrix looks like:
> > >
> > > $$\begin{pmatrix}1&a& a^2&\cdots &a^{N-1}\\\\
> > > a&1&a&\cdots & a^{N-2}\\\\
> > > \vdots&\vdots&\ddots&&\vdots\\\\
> > > a^{N-1}&a^{N-2}&\cdots&&1\\\\
> > > \end{pmatrix}$$
> > >
> > > The determinant of this matrix can be calculated analytically as $(1-a^2)^{N-1}$ (proof in [this stack exchange answer](https://math.stackexchange.com/questions/2578797/how-to-compute-the-determinant-of-this-toeplitz-matrix) from a few years ago). In particular, this value is non-zero for all valid $a,N$, implying this matrix is full rank. Setting $N=M+1$ contradicts the assumption of an $M$-dimensional feature map, and repeating this argument for all finite $M$ proves that no finite-dimensional feature map can exist, _thereby completing our proof_.
> > >
> > > We conjecture that $T_{DP}$ has the same property. The kernel matrix for $T_{DP}$ on these same set of points also "looks like" the identity, but it does not have Toeplitz structure so we could not analytically calculate its determinant. Numerically it seems to have non-zero determinant for large $N$ though.

---

> > > > ### Comment · Reviewer_stq5 · 2023-08-14
> > > > **resp**
> > > >
> > > > Thanks for the response. I agree that Tanimoto is indeed full-rank, and thus random features are useful to derive. I'm also happy with the clarity improvements, although there are quite a lot of changes for camera ready.
> > > >
> > > > I'm raising my score to 5.

---

### Author Rebuttal · Authors · 2023-08-09

We would like to thank all reviewers for their constructive comments. We have left a response to each reviewer answering their questions, but would like to provide a summary here.

First, we appreciate that many reviewers complimented various aspects of our work including:

- Theoretical analysis and technical contribution (pMsa, jyA1, A9mq, b6mP)
- Clarity (b6mP, A9mq, jyA1)
- Honesty about limitations (A9mq, jyA1)

The main concerns raised by the reviewers were:
- Notation (stq5): there was some ambiguity in subscripts and instances where we defined probability distributions implicitly. _We will change this in the next version of the manuscript._
- Status of theorem 4.7 (b6mP, pMsa): _this is a novel theorem, and we are certain about its correctness_. Cohen et al (2015) pioneered the analysis technique and set a reference for what a “good” error bound is in these types of problems, and our comment in the discussion was about the bound not being tight or optimal, _not_ a statement that we are unsure about its correctness.
- Relationship to deep learning methods (A9mq, jyA1): we believe and clearly acknowledge in the paper that there are circumstances where deep learning methods perform much better than kernel methods: for example on problems with large datasets like in section 5.3. However, kernel methods have distinct advantages on small datasets or in instances where robustness / interpretability are important. The Thompson sampling experiment in 5.2 is a typical setting where we envision our method will be used: even with a small number of training data points, sampling scales poorly with the number of _evaluation_ points but can be greatly accelerated using our random features (at the cost of some approximation error).

  Furthermore, because the kernels studied in this work are differentiable and valid for continuous inputs, they could be combined with learned features in the future, for example via “Deep Kernel Learning” (Wilson et al 2016). We leave this to future work.

We hope we have adequately addressed the concerns of all reviewers and appreciate the time that you have put into the reviewing process. We would be happy to answer any further questions.

---

> ### Comment · Area_Chair_EWZF · 2023-08-18
> **Rebuttal**
>
> Thank you for your rebuttal. We will take it into account in making the final recommendation.

---

### Decision · Program_Chairs · 2023-09-21

**Decision:**

Accept (poster)

**Comment:**

The paper focuses on computing efficient approximations to the Tanimoto coefficient, an important similarity measure in many applications, especially in applications on molecules. The paper puts forward two different random feature based approximations of the Tanimoto kernel, and analyze their theoretical properties. The methods are tested empirically in extensive experiments that showed the practical relevance of the method. The reviewers found the paper technically solid. The minor concerns of the reviewers were adequately addressed in the rebuttals and the following discussion.